# Tailoring grain boundary stability of zinc-titanium alloy for long-lasting aqueous zinc batteries

Yunxiang Zhao[1], Shan Guo[1], Manjing Chen[1], Bingan Lu [2], Xiaotan Zhang[1] ✉, Shuquan Liang[1] ✉ & Jiang Zhou [1] ✉

The detrimental parasitic reactions and uncontrolled deposition behavior derived from inherently unstable interface have largely impeded the practical application of aqueous zinc batteries. So far, tremendous efforts have been devoted to tailoring interfaces, while stabilization of grain boundaries has received less attention. Here, we demonstrate that preferential distribution of intermetallic compounds at grain boundaries via an alloying strategy can substantially suppress intergranular corrosion. In-depth morphology analysis reveals their thermodynamic stability, ensuring sustainable potency. Furthermore, the hybrid nucleation and growth mode resulting from reduced Gibbs free energy contributes to the spatially uniform distribution of Zn nuclei, promoting the dense Zn deposition. These integrated merits enable a high Zn reversibility of 99.85% for over 4000 cycles, steady charge-discharge at $10\,mA\,cm^{-2}$, and impressive cyclability for roughly 3500 cycles in Zn-Ti// $NH_4V_4O_{10}$ full cell. Notably, the multi-layer pouch cell of 34 mAh maintains stable cycling for 500 cycles. This work highlights a fundamental understanding of microstructure and motivates the precise tuning of grain boundary characteristics to achieve highly reversible Zn anodes.

The integration of intermittent power supply from renewable resources is of great significance to the impending energy transition, given the looming concerns about environmental deterioration and energy crisis[1–3]. Rechargeable batteries based on zinc (Zn) chemistry are an in-principle promising energy storage technology, owing to their modularity, adaptability to grid systems, and favorable comparison with other prevailing storage technologies[4]. The optimism stems from their inherent benefits, such as safety, environmental benignity, ease of manufacture, and, most significantly, low cost arising from their natural abundance, rendering them highly desirable for large-scale grid storage systems[5].

Notwithstanding these advantages, practical implementation of aqueous zinc batteries (AZBs) is impeded by the thermodynamically favored hydrogen evolution reactions (HER) and corrosion[6,7]. Zn

electrodeposition in irregular, non-planar morphologies presents another hurdle, leading to limited reversibility and insufficient utilization of the electrochemically active Zn anode[8,9]. To address these issues, suppressing side reactions and regulating Zn deposition behavior are critical for the prolonged battery lifespan[10]. While surface modification and substrate exploration are widely reported as external strategies, alloying has emerged as an effective means of tailoring physicochemical properties through self-reconstruction[11]. Importantly, the incorporation of other metallic elements into the Zn matrix enables precise control over Zn nucleation and growth behavior within the alloy anode, while significantly enhancing its corrosion resistance[12,13]. The chemical component, phase composition (solid solutions and intermetallic compounds, IMCs), and microstructure (phase distribution, grain structure, etc.) have a significant impact on

[1]School of Materials Science and Engineering, Hunan Provincial Key Laboratory of Electronic Packaging and Advanced Functional Materials, Central South University, Changsha 410083 Hunan, China. [2]School of Physics and Electronics, Hunan University, Changsha 410082 Hunan, China. ✉ e-mail: zhangxiaotan@csu.edu.cn; lsq@csu.edu.cn; zhou_jiang@csu.edu.cn

the properties of Zn alloys[14]. Various inspiring schemes, including alloying with high HER overpotential metals[15–17], constructing zincophilic interfaces[18,19], and forming electrostatic shielding protective layers[20], have been developed. Nevertheless, a profound understanding of the chemistry behind microstructure features remains in its infancy.

Recently, an increasing number of researchers have been devoted to addressing the challenges associated with utilizing Zn anodes in mildly acidic aqueous electrolytes[21]. Typically, Zn and its alloys exhibit both general corrosion and localized corrosion, with the latter being more destructive[22]. From a metallographic perspective, the microstructure features of metal substrates play a pivotal role in the formation of galvanic cells that drive localized corrosion process[23]. Grain boundaries (GBs), as essential structural units connecting grains in polycrystalline metals, exhibit enhanced reactivity compared to crystallographic planes[24]. Consequently, localized corrosion tends to initiate at GBs with high corrosion susceptibility and then propagate towards the interior of grains or advance to the depth, resulting in severe intergranular corrosion[25,26]. Therefore, understanding the thermodynamic-driven corrosion process at the microscopic level is urgent, and modifying GBs emerges as a new approach to tailor metal anode. Despite the encouraging progress in state-of-the-art lithium alloy anodes involving the modulation of GB characteristics[27,28], grain boundary engineering (GBE) has not been extensively studied in AZBs.

On the other hand, the overall morphology of Zn anode is frequently employed as a qualitative indicator of improved deposition, yet the initial nucleation and growth processes are generally ignored, despite their equal significance[29]. In terms of electrodeposition kinetics, homogeneous nucleation guides the subsequent growth process, which is critical for achieving consistently uniform Zn deposition, thus ensuring the long-term cyclic stability of full cells[30]. Additionally, the evenly distributed Zn nuclei can effectively alleviate the "tip effect" by interacting with the electric field and ion flux distribution[31]. Developing a novel Zn alloy that offers exceptional corrosion resistance and well-controlled Zn deposition to achieve practical AZBs is highly desirable but remains a significant challenge. As a proof of concept, titanium (Ti) is selected as the alloying element based on the following considerations: (i) Ti inherently possesses high corrosion resistance, biocompatibility (non-toxicity), and the potential to improve processing characteristics, favoring mass production[32]. (ii) Zn-Ti binary system exhibits limited solubility of Ti in Zn, indicating the possibility of compound formation or GB segregation[33]. (iii) Ti-containing IMCs preferentially distribute at GBs, strengthening the bonding force and blocking the corrosion path[34,35]. (iv) In comparison to reported Zn alloys with high levels of inactive components (e.g., $Zn_3Hg$)[36], a low content of Ti maximizes the utilization of Zn anode.

Here, we delve into the correlation between surface microstructure and corrosion behavior, and introduce an innovative concept of GBE (Fig. 1a) to stabilize Zn anode while simultaneously regulating Zn deposition behavior. Corrosion reactions tend to occur at GBs due to their elevated reactivity. To inhibit intergranular corrosion, we propose modifying the GBs of metallic Zn by constructing a Zn-Ti dual-phase alloy, where Ti-containing IMCs are formed and thermodynamically stabilized at GBs. Detailed electrochemical tests and immersion experiments validate the substantial suppression of HER-induced corrosion, regardless of aging or extended cycling. Moreover, the Ti-containing IMCs with high zincophilicity endow Zn-Ti alloy with diminished nucleation energy barrier, promoting favorable nucleation mode, and the superiority in subsequent growth process is corroborated through finite element simulations. As validation, half cell with Zn-Ti alloy delivers an impressive coulombic efficiency (CE) of 99.85% over 4000 cycles at 5 mA cm$^{-2}$/1 mAh cm$^{-2}$ and long-term cycling stability of up to 1100 h at 2 mA cm$^{-2}$/2 mAh cm$^{-2}$. In particular, the scaled-up pouch cell paired with $NH_4V_4O_{10}$ cathode accomplishes the cyclability of 85% capacity retention over 500 cycles. Our discovery specifies a rarely addressed perspective concerning microstructure features of polycrystalline Zn and may invoke a paradigm shift to future alloy anode design.

## Results

### Fabrication process and microstructure features of Zn-Ti alloy

A conventional metallurgical process has been developed for the construction of Zn-Ti alloy (Supplementary Fig. 1). Metallic Zn was molten in the crucible in advance, and an appropriate amount of Ti was introduced. The ingot was then processed into desired sizes with controllable thickness via mechanical rolling, which favors mass

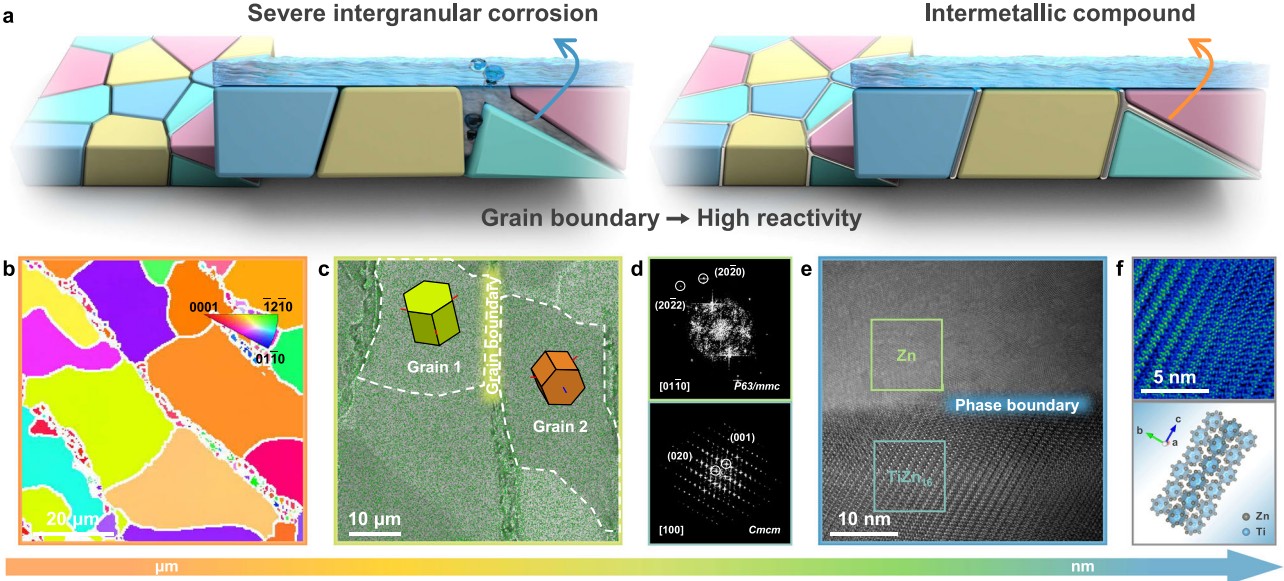

**Fig. 1 | Structural characterizations of Zn-Ti alloy. a** Schematic illustration of GBE. **b** EBSD IPF mapping of Zn-Ti alloy. **c** SEM-EDS mapping of Zn-Ti alloy. The typical grain boundary structure separates Grain 1 and Grain 2, and the geometries visually display the crystal orientations of these two grains. **d** FFT patterns of the selected areas in HR-TEM image. **e** HR-TEM image of Zn-Ti alloy. A distinct phase boundary distinguishes the Zn phase and the $TiZn_{16}$ phase. **f** Magnified $TiZn_{16}$ region and corresponding crystal structure.

production, after undergoing a multi-step annealing procedure. As illustrated in the Zn-Ti binary phase diagram, the solid solubility of Ti in Zn is extremely limited, and IMCs composed of $TiZn_x$ can be formed according to their proportions (Supplementary Fig. 2). It is recognized that IMCs tend to segregate at crystal defects such as GBs, thereby offering the potential to lower the reactivity of GBs and ultimately minimize the distortion energy of the entire system[27,37]. Here, we focus on the alloy containing as little as 0.5 wt% Ti (Supplementary Table 1), which predictably yields a Zn-rich solid solution and $TiZn_{16}$ IMCs. Supplementary Fig. 3 shows the typical X-ray diffraction (XRD) patterns of Zn-Ti alloy, with the predominant peaks corresponding to the hexagonal Zn phase, apart from the weak ones attributed to the orthorhombic $TiZn_{16}$ phase. Subsequently, multi-scale characterizations were employed to investigate the microstructure features of Zn-Ti alloy. Prior to this, the sample was subjected to a twin-jet electro-polishing procedure to eliminate the influence of surface impurities and roughness on electronic signals. The electron backscatter diffraction (EBSD) inverse pole figure (IPF) mapping reveals that the Zn-Ti alloy comprises multiple grains exhibiting diverse orientations, with "zero solution" pixels primarily distributed along the GBs (Fig. 1b). Additionally, scanning electron microscope (SEM) backscattered electron image, accompanied by energy disperse spectroscopy (EDS) elemental mapping obtained from the same region, indicates that most Ti concentrates at the GBs (Fig. 1c and Supplementary Fig. 4). High-resolution transmission electron microscope (HR-TEM) and corresponding EDS mapping images demonstrate the well-defined interface and symbiotic Zn and $TiZn_{16}$ IMCs (Fig. 1e and Supplementary Fig. 5). The fast Fourier transform (FFT) patterns of selected areas further confirm the distinct separation of orthorhombic $TiZn_{16}$ phase and hexagonal Zn phase during the solidification process (Fig. 1d and Supplementary Fig. 6). In the magnified view of $TiZn_{16}$ region, distinct lattice fringes and compacted atomic arrangements are evident, consistent with the sophisticated crystal structure (Fig. 1f). Therefore, a Zn-Ti dual-phase alloy with GB segregation has been successfully fabricated and subsequently utilized for corrosion resistance assessment.

## Corrosion resistance via grain boundary engineering

The corrosion behavior of Zn-Ti alloy, with bare Zn as a control, was investigated through electrochemical tests and immersion experiments. The suppressed Zn corrosion was initially evidenced by Tafel polarization curves in the three-electrode system, in which the corrosion current density of bare Zn (3.396 mA cm$^{-2}$) corresponds to 184% of the value obtained for Zn-Ti alloy (1.842 mA cm$^{-2}$) (Fig. 2a). The decrease in corrosion current density suggests its diminished corrosion rate, while the observed variation in corrosion potential can be deemed negligible. Typically, Zn corrosion under acidic conditions proceeds via cathodic control, and the kinetics of HER dictates the overall corrosion rate[38]. Therefore, HER activity was assessed using linear sweep voltammetry (LSV) measurements, and the lower onset HER potential for Zn-Ti alloy manifests a reduced susceptibility to corrosion, potentially attributed to GBE (Supplementary Fig. 7).

The propensity of Zn anodes to degrade under prolonged aging and cycling in mildly acidic electrolytes, owing to reactions with water solvent, results in the irreversible consumption of metallic Zn into by-products[39]. To evaluate the shelf life, Zn anodes were immersed in $ZnSO_4$ electrolyte for varying durations. It is evident that for bare Zn, the diffraction peaks corresponding to $Zn_4SO_4(OH)_6 \cdot 3H_2O$ (PDF#39-0689) begin to appear after 1 day of immersion and become significant after 10 days, indicating severe corrosion reactions (Fig. 2b). The corresponding SEM and EDS mapping further corroborate this result, revealing discernible flake-like by-products and noticeable signs of corrosion (Supplementary Fig. 8). Conversely, the surface of Zn-Ti alloy displays minimal by-products, with the diffraction peaks of zinc hydroxysulfate remaining inconspicuous even after 10 days of immersion, demonstrating the excellent corrosion inhibition

capability. Post-mortem analysis was then performed to validate its stability upon long cycles. The small-sized and dense-stacked morphology is realized by Zn-Ti alloy after 50 cycles, whereas the notorious by-products are found interwoven within the Zn deposits on bare Zn (Supplementary Fig. 9). For cross-sectional observations, bare Zn exposes a rough surface (thickness varies from 5 to 14 μm) featuring obvious protrusions accumulating loosely, likely to develop into dendrites and trigger short circuit (Fig. 2c)[40]. In stark contrast, Zn-Ti alloy yields a well-preserved and integrated Zn deposition layer (thickness of about 8 μm), with the signals of by-products nearly undetectable derived from the effective suppression of HER. This highly compact Zn deposition behavior is pivotal in mitigating side reactions, thus promising an extended lifespan. It is worth noting that the peaks corresponding to $TiZn_{16}$ IMCs retain their intensity regardless of aging or extended cycling, suggesting a sustainable inhibitory effect independent of Zn migration. Its thermodynamic stability at GBs was also verified from the pure curves in the cyclic voltammetry (CV) testing (Supplementary Fig. 10).

Diagnosing the underlying degradation mechanism is essential for elucidating the enhanced corrosion resistance capability. As mentioned earlier, metallic Zn is thermodynamically active towards aqueous electrolytes, rendering general corrosion inevitable (Fig. 2d). Nonetheless, since it acts evenly across the entire surface, its effect is not that severe, at least gradually[41]. The presence of GBs with higher energy relative to the grain surface tends to cause corrosion to propagate along them, where the micro-galvanic effect exacerbates the localized corrosion (Fig. 2e)[42]. To further clarify the significance of GBE in inhibiting intergranular corrosion, EBSD analysis was performed to monitor the reaction process, in which the phase recognition was quantified on the basis of image identification. As shown in Fig. 2f, the scanning pattern with color mapping is identified as active Zn, whereas the white parts correspond to the unrecognizable phases that can be attributed to $TiZn_{16}$ IMCs or corrosion pits. To discern between these phases, additional SEM images were employed, with the former revealing a continuous distribution (Supplementary Fig. 11). The EBSD IPF mapping of both bare Zn and Zn-Ti alloy is clear at pristine state. Afterwards, the bare Zn surface displays a typical localized corrosion morphology. The most pronounced corrosion damage manifests along the GBs, followed by its expansion towards the interior of grains or even deeper regions, as indicated by arrows. In contrast, minimal signs of corrosion are observed on Zn-Ti alloy, as further confirmed by the ratio of phase recognition, which only marginally decreases from 98% to 95% after immersion (Fig. 2g). However, the recognizable phase of bare Zn decreases sharply from nearly 100% to 89%. The distinct corrosion morphology was then verified through confocal laser scanning microscope (CLSM) imaging, where irregular pits with depths ranging from 1 to 2 μm are discernible on the surface of bare Zn, signifying the initiation of micro-pitting corrosion (Fig. 2h). Conversely, owing to the suppressed localized corrosion, Zn-Ti alloy retains a relatively flat appearance with low roughness (Fig. 2i). Clearly, the parasitic reactions mentioned above do affect the reversibility of Zn plating/stripping in actual cells, and the proposed "reservoir" protocol was then leveraged to quantify the amount of Zn consumed[6]. A high average CE of 98.49% can be achieved for Zn-Ti alloy, whereas bare Zn exhibits erratic voltage behavior induced by corrosion reactions and is unable to complete the protocol (Fig. 2j). These results collectively underscore that the remarkable corrosion resistance of Zn-Ti alloy can be attributed to the substantial suppression of intergranular reactions.

## Favorable nucleation and growth models

Obviously, the nature of metallic substrate has a profound influence on Zn deposition, prompting the question of how Zn-Ti alloy interacts with and regulates the deposition behavior. The morphological evolution of Zn deposits over time was firstly examined by ex-situ SEM

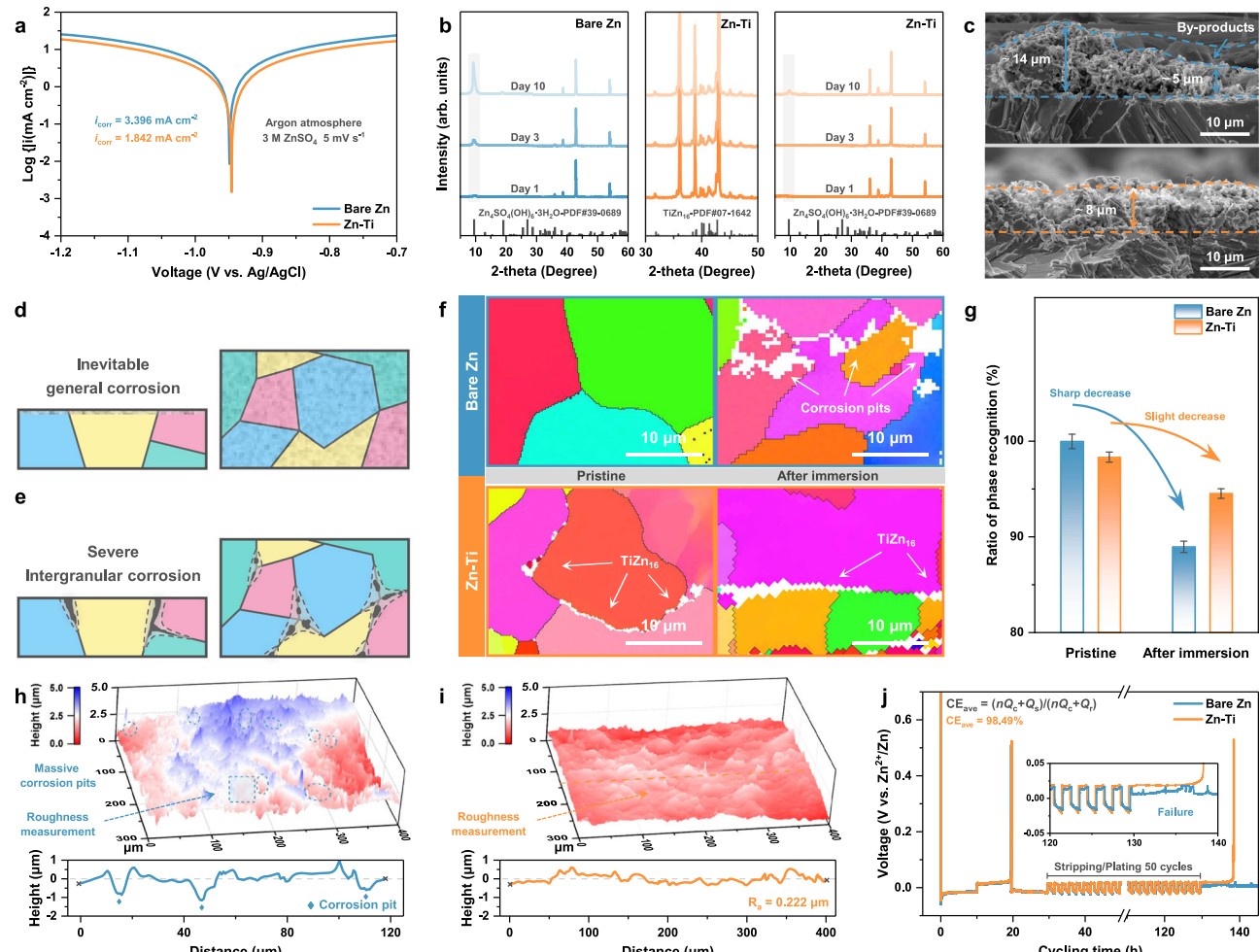

**Fig. 2 | Suppression of Parasitic Reaction through Grain Boundary Engineering.**
**a** Tafel plots. **b** Ex-situ XRD patterns of Zn anodes after immersion in ZnSO₄ electrolyte for different days. **c** Cross-sectional SEM images of Zn anodes after 50 cycles. Schematic diagram of **d** general corrosion and **e** intergranular corrosion.
**f** EBSD IPF mapping images of Zn anodes at pristine state and after 24 h of immersion. Different colors represent distinct crystal orientations, while the white parts correspond to unrecognizable phases, attributable to corrosion pits or $TiZn_{16}$ IMCs. **g** Corresponding ratio of phase recognition based on image identification. The error bars indicate the mean absolute deviation. CLSM imaging of corrosion morphologies on **h** bare Zn and **i** Zn-Ti alloy, and corresponding surface profiles extracted along the location marked. **j** "Reservoir" protocol for evaluating average Zn plating/stripping CE.

imaging. The initial state of Zn deposition is depicted in Fig. 3a, and it is noteworthy that the deposits obtained in a short time are primarily determined by the nucleation process. It can be observed that $Zn^{2+}$ tends to nucleate randomly on bare Zn. With the proceeding of deposition, these sparse nuclei continue to grow and coalesce, resulting in pronounced protrusions and leaving behind widely uncovered regions. Subsequent $Zn^{2+}$ deposition on existing Zn nuclei induces inhomogeneity in the growth process, which can be traced back to the uneven distribution of nucleation sites, consequently intensifying the inclination towards localized Zn deposition (Fig. 3b). Moreover, the morphology of Zn deposits transforms from blocky to mossy, potentially aggravating the corrosion reactions (Supplementary Fig. 12)[43]. In contrast, the deposits on Zn-Ti alloy exhibit a particulate-like morphology, as revealed by time-series SEM images. Similar-sized Zn particles nucleate uniformly at the initial stage of deposition. As the deposition progresses, the size of Zn nuclei increases, along with a rise in their quantity, suggesting a time-dependent progressive feature (Supplementary Fig. 13). The abundant Zn nuclei guide the subsequent growth process, and when adjacent nuclei come into contact with each other, they gradually merge into a dense Zn deposition layer that covers the whole surface of the substrate. The height characteristics of Zn deposits were then examined

using atomic force microscopy (AFM). Notably, even after deposition for 60 min, the upper surface of Zn-Ti alloy consistently maintains a low roughness, while the deposits on bare Zn display an undulating topography with distinct slopes whose height difference exceeds 3.8 μm (Fig. 3c). Evidently, the Zn deposition on Zn-Ti alloy reveals a homogeneous particulate-like deposition mode, quite different from the uneven dendrite-like deposition mode observed on bare Zn. Of particular note is the evident differences between the two morphologies right from the early stage of deposition, implying the existence of a critical factor in the nucleation process that distinguishes the deposition modes. In addition, the stripping process following plating was also investigated, which holds equal importance[44]. In contrast to the uniformly stripped surface observed on Zn-Ti alloy, loosely distributed "dead Zn" and deep pits are distributed on the surface of bare Zn (Supplementary Fig. 14). In fact, the inhomogeneous plating/stripping behavior will intertwine and recur over extended cycles, ultimately leading to interfacial instability and a decline in battery performance. These findings emphasize the significance of initial deposition quality for the subsequent plating/stripping reversibility, as discussed below.

According to classical nucleation theory, the formation of a new phase needs to overcome an energy barrier, which can be reflected

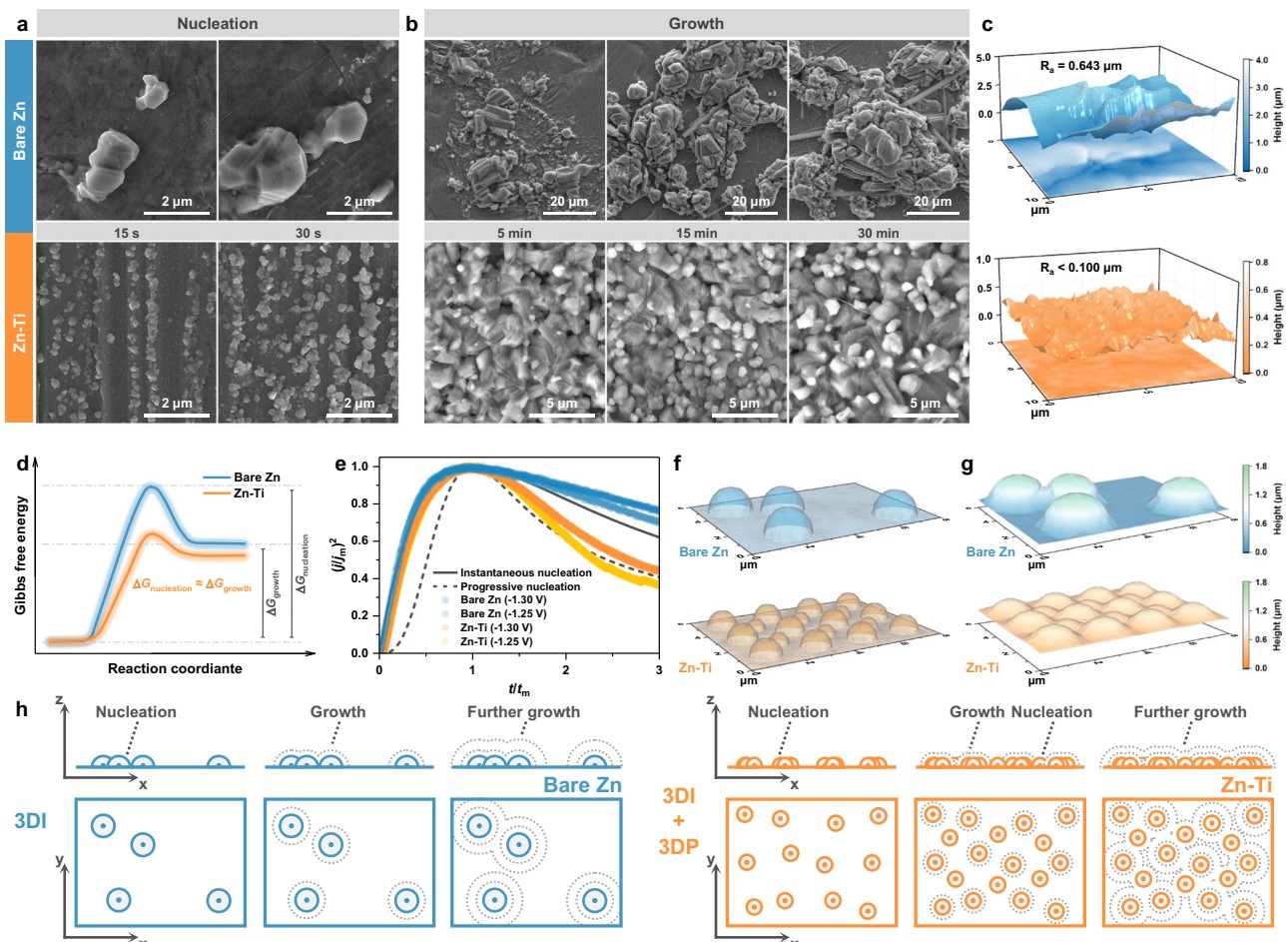

**Fig. 3 | Mechanisms of Zn nucleation and growth.** Time-series SEM images of Zn **a** nucleation and **b** growth. **c** AFM height images after Zn deposition for 60 min. **d** Semi-quantitative description of Gibbs free energy during Zn deposition. **e** Experimental dimensionless transients in comparison with the theoretical 3D nucleation models. Simulated **f** initial nuclei models and **g** final Zn deposition models after 240 s of plating. **h** Schematic illustration of the Zn nucleation and growth mechanisms.

electrochemically[45]. Specifically, a detailed examination of the characteristic overpotential parameters will enhance the understanding of this thermodynamically related process, as depicted in Supplementary Fig. 15. The voltage profile obtained during galvanostatic deposition reveals an apparent voltage dip at the onset of electrodeposition on bare Zn, measuring 51 mV. In contrast, the Zn-Ti alloy displays a lower nucleation overpotential of only 24 mV, indicative of a reduced energy barrier. Note that there are minimal disparities in the plateau overpotentials, suggesting that the substrate primarily influences nucleation rather than growth process[46]. The significant differences in the nucleation overpotentials are thought to be associated with the distinct affinities toward Zn[47]. To gain a deeper insight into zincophilic behavior, the binding energies of Zn atom attached to the substrates were evaluated using density functional theory (DFT) calculations, with the selection of crystal planes guided by XRD results. As summarized in Supplementary Fig. 16, the calculated binding energy values between Zn atom and $TiZn_{16}$ (241) facet, (134) facet, and (313) facet are −1.07, −1.71, and −1.20 eV, respectively, which are more negative than those between Zn atom and Zn facets (−0.26, −0.66 and −0.85 eV). This indicates that $TiZn_{16}$ IMCs could potentially serve as nucleation sites, which endow the Zn-Ti alloy with high zincophilicity to facilitate Zn deposition, thus leading to a reduced nucleation overpotential and a lower energy barrier. Another noteworthy observation is that the voltage profile of Zn-Ti alloy exhibits a flat voltage turning without any noticeable voltage spike, which contrasts sharply with the response

observed on bare Zn. Typically, the difference between two characteristic overpotentials is regarded as an indicator for assessing the gap of driving forces between nucleation and growth processes[48]. As semi-quantitatively shown in Fig. 3d, the large energy difference between the Gibbs free energy for Zn nucleation ($\Delta G_{nucleation}$) and growth ($\Delta G_{growth}$) on bare Zn partly restricts nucleation while favoring the growth process. As a result, $Zn^{2+}$ tends to deposit onto existing Zn nuclei, which may develop into dendrites as capacity increases[49]. In contrast, the $TiZn_{16}$ IMCs with high zincophilicity reduce the $\Delta G_{nucleation}$ to the same level of $\Delta G_{growth}$, thereby promoting more and denser Zn nuclei and resulting in homogeneous Zn deposition.

Since the electrocrystallization of new phases is a strong function of overpotentials and can be directly reflected by current transients, chronoamperometry (CA) was employed to further probe the kinetics of Zn nucleation and growth. All the transients exhibit a peak current ($j_m$), indicating the typical three-dimensional (3D) nucleation process with hemispherical diffusion control (Supplementary Fig. 17)[50]. Initially, the rise in current reflects the expansion of electroactive area, owing to the growth in size of existing nuclei and the generation of new nuclei on the surface[51]. Subsequently, the current decreases and approaches a steady state due to the overlapping of neighboring nuclei and their diffusion zones, indicating a diffusion-controlled growth process[52]. It is evident that the Zn-Ti alloy demonstrates a slower and more progressive nucleation process in comparison to bare Zn, as substantiated by the extended time required to reach the peak current

$(t_m)$. Furthermore, the current-time transients were normalized based on $j_m$ and corresponding $t_m$, and compared with the Scharifker-Hills model[52]. Based on this theory, nucleation occurs in two modes, instantaneous (I) or progressive (P), depending on whether new nuclei emerge instantaneously at the beginning or progressively over time, and the mathematical expressions are described by:

$$3DI : \left(\frac{j}{j_m}\right)^2 = 1.9542\left(\frac{t}{t_m}\right)^{-1}\left\{1 - \exp\left[-1.2564\left(\frac{t}{t_m}\right)\right]\right\}^2 \qquad (1)$$

$$3DP : \left(\frac{j}{j_m}\right)^2 = 1.2254\left(\frac{t}{t_m}\right)^{-1}\left\{1 - \exp\left[-2.3367\left(\frac{t}{t_m}\right)^2\right]\right\}^2 \qquad (2)$$

The normalized dimensionless transients are shown in Fig. 3e, revealing that the nucleation on bare Zn follows closely the response predicted for 3DI, that is, the nucleation sites are relatively limited and become exhausted at early stage of the process. Conversely, the nucleation behavior observed on Zn-Ti alloy follows somewhat a hybrid model, that is, 3DI for $t < t_m$ as well as 3DP and diffusion-controlled growth for $t > t_m$. In this scenario, nucleation sites are progressively activated and the process is accompanied with nuclei growth, resulting in smaller particle sizes typically. These model results coincide well with the morphological differences observed during initial Zn deposition. The distinctions in nucleation modes will inevitably exert the localized electrochemical environment, thus shaping the subsequent growth behavior.

Additionally, finite element simulations using COMSOL Multiphysics were conducted to further understand the post-nucleation growth process. Hemispheres were employed to mimic Zn nuclei structure based on the preceding SEM observations. The initial modeling, depicted in Fig. 3f, illustrates that Zn nuclei of varying sizes are evenly distributed on Zn-Ti alloy, while block-like deposits are randomly scattered on bare Zn. Initially, the electrolyte current exhibits a discernible inclination to flow towards the Zn nuclei, with this tendency being more pronounced for bare Zn due to limited nucleation sites (Supplementary Fig. 18). Upon extending the simulation time, $Zn^{2+}$ accumulates onto existing Zn nuclei along the trajectory defined by current vector, intensifying the propensity for localized deposition and consequently resulting in prominent protrusions (Fig. 3g and Supplementary Fig. 19). In contrast, the abundant Zn nuclei present on Zn-Ti alloy interact with the electric field and ion flux, diminishing the inhomogeneity of Zn growth. As deposition progresses, the gaps between Zn nuclei are gradually filled and the surface becomes smoother, in accordance with the experimental findings.

Building on the aforementioned observations and analysis, the potential nucleation and growth mechanisms were schematically summarized in Fig. 3h. For bare Zn following the 3DI model, Zn nuclei emerge randomly on the limited nucleation sites within a short time, and they tend to grow in their former positions without the formation of new nuclei due to the elevated nucleation barrier. The subsequent growth process is driven by the 3D volumetric extension of such structure, intensifying the heterogeneity of deposition, and thus the final appearance exhibits pronounced protrusions. For Zn-Ti alloy following the 3DI + 3DP model, $TiZn_{16}$ IMCs with high zincophilicity could provide more nucleation sites that undergo progressive activation. The evenly distributed Zn nuclei guide the further growth, thereby yielding the relatively flat Zn deposition layer. In brief, Zn-Ti alloy was found to be effective in promoting favorable nucleation and growth processes, which are crucial for highly reversible AZBs.

## Electrochemical performance of half cells and full cells

To validate the merits of Zn-Ti alloy for reversible Zn migration, plating/stripping measurements were performed in both symmetric and asymmetric cells. As expected, the symmetric cell using Zn-Ti alloy maintains a stable and low voltage hysteresis of 27 mV at 1 mA cm$^{-2}$/ 1 mAh cm$^{-2}$, resulting from the reduced energy barrier (Supplementary Fig. 20a). Conversely, the profile for bare Zn exhibits erratic fluctuations after 250 h, accompanied by a noticeable rise in voltage hysteresis, which is a common signature of HER-induced corrosion[53]. As the areal capacity increases to 2 mAh cm$^{-2}$, a target value for commercial AZBs[7], the cell with bare Zn encounters difficulty in sustaining cycling for over 180 h (Fig. 4a). The voltage response approximates the square-wave shape with an extremely low voltage hysteresis, indicative of the internal short circuit caused by dendrite growth[54]. In comparison, the Zn-Ti symmetric cell demonstrates exceptional stability for more than 1100 h, without the widely reported "soft shorts" circumstance, as evidenced by the well-maintained slope curves throughout the cycling process. The evolution of Nyquist plots revealed their interfacial stability, as shown in Supplementary Fig. 21. The Zn-Ti symmetric cell exhibits small and stable interfacial impedance, consistent with the enhanced reaction kinetics and well-preserved plating/stripping morphology (insets of Fig. 4a). In contrast, the pronounced mossy Zn and deep pits on bare Zn upon prolonged cycling exacerbate the side reactions due to increased surface area exposure, which accounts for the escalating charge transfer resistance. For further evidence, operando optical microscopy with a transparent window was employed to visually monitor the Zn plating/stripping behavior. A precise formation of Zn dendrites can be observed on bare Zn at 15 min, relative to the dendrite-free planar Zn deposition layer on Zn-Ti alloy throughout the galvanizing process (Supplementary Fig. 22). Analyzing the rate capability of Zn-Ti alloy by stepwise increasing the current density discloses the low and steady voltage hysteresis, profiting from the favorable $Zn^{2+}$ migration kinetics and superior interfacial stability (Fig. 4b and Supplementary Fig. 23). By comparison, cells with bare Zn display voltage fluctuation or even short-circuit under high current densities. To obtain a deeper comprehension of the accelerated charge transfer, impedance measurements were conducted at incremental temperatures. Notably, Zn-Ti alloy exhibits consistently lower interfacial impedance across all temperature ranges (Supplementary Fig. 24). Accordingly, the activation energy was calculated using the Arrhenius equation, which reveals a reduction in activation energy from 46.05 kJ mol$^{-1}$ to 39.84 kJ mol$^{-1}$ when employing Zn-Ti alloy instead of bare Zn, confirming the improved reaction kinetics. The feasibility of Zn-Ti alloy for high-power application was then validated under harsh conditions (Supplementary Fig. 20b). Impressively, even at 8.85 mA cm$^{-2}$ and 25% depth of discharge, the symmetric cell with Zn-Ti alloy maintains a steady cycle for 120 h, nearly twice as that with bare Zn, indicating its potential for practical applications (Supplementary Fig. 25). The reversibility of Zn-Ti alloy was further investigated using an asymmetric cell configuration, as depicted in Fig. 4c. The Zn plating/stripping behavior on copper (Cu) substrate initially undergoes a lattice matching process, after which it stabilizes at over 99.60% for 4000 cycles. This results in a high average CE of 99.85% for a total cumulative capacity of 4 Ah cm$^{-2}$. To the best of our knowledge, this is almost the longest cycle life and highest cumulative capacity achieved at such high CE and current density. Surprisingly, the average plating/stripping CE even reaches 99.90% at selected cycles (inset of Fig. 4c), reflecting the close-to-full control of parasitic reactions. Moreover, the Zn-Ti alloy consistently displays lower and more stable voltage hysteresis throughout the cycling process (Fig. 4d and Supplementary Fig. 26). To fully explore the commercial viability of Zn-Ti alloy, its cyclability and CE were also examined under different current densities and areal capacities (Supplementary Fig. 27). Regardless of the difference in test conditions, asymmetric cells with bare Zn suffer inferior cycling performance, consequently falling into fast decay. The

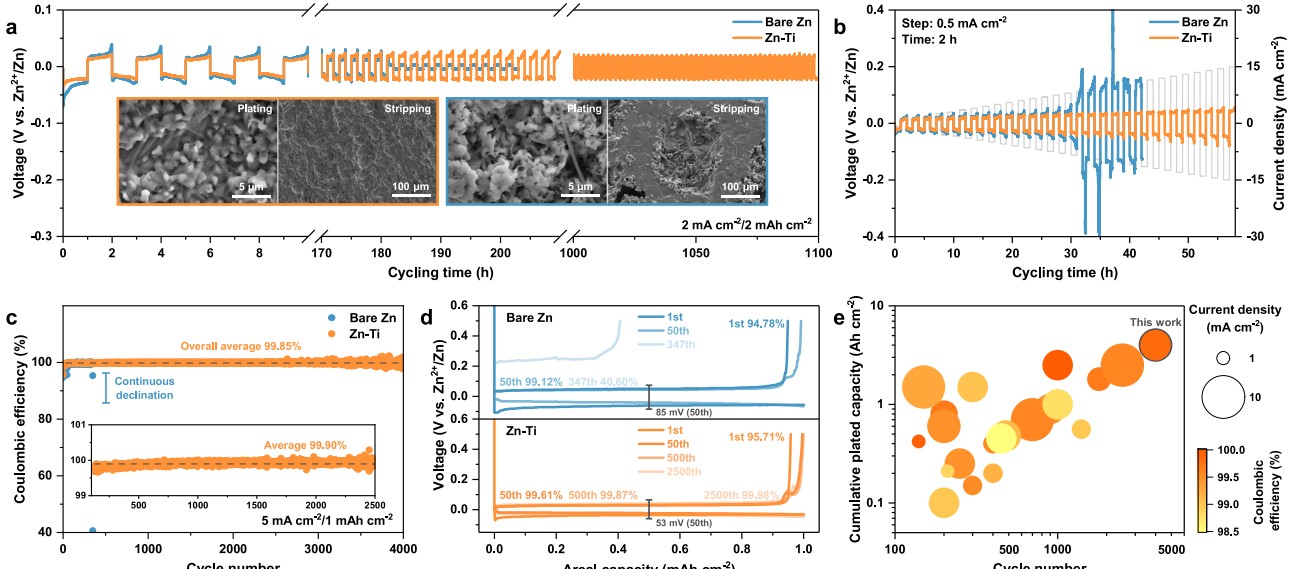

**Fig. 4 | Reversible Zn plating/stripping behavior. a** Cyclic stability of Zn//Zn symmetric cells at 2 mA cm$^{-2}$/2 mAh cm$^{-2}$ (insets are SEM images of Zn anodes after 75 cycles). **b** Voltage evolution of Zn//Zn symmetric cells at step-increased current densities. The **c** plating/stripping CE (inset is magnified view of selected cycles between the 100th and 2500th cycle) and corresponding **d** voltage profiles of Zn// Cu asymmetric cells at 5 mA cm$^{-2}$/1 mAh cm$^{-2}$. **e** A comparison of Zn plating/stripping performance in recent demonstrations with our work.

drastically fluctuating signals before cell failure originate from the random reconnection of inactive Zn, which is closely related to the nonuniform deposition[55]. Overall, the superior electrochemical performance of Zn-Ti alloy can be attributed to the integrated merits of favorable deposition modes and corrosion resistance derived from GBE. In terms of four critical parameters, the metrics in this work are comparable to or better than previously reported Zn anodes, including Zn@IPLs, Zn@OPLs, Zn@OIPLs and other ZAs (Fig. 4e and Supplementary Table 2). These prospective results provide support for using Zn-Ti alloy as opposed to conventional Zn anode in highly reversible AZBs.

The electrochemistry of Zn-Ti alloy was evaluated in full cells, using $NH_4V_4O_{10}$ as a representative cathode, to probe its potential for actual applications. In the form of coin cell, the battery with bare Zn fails rapidly after 263 cycles at 1 A g$^{-1}$ under the benchmark of 80% capacity retention, due to the increased polarization (Supplementary Fig. 28). By contrast, the cycle life of battery with Zn-Ti alloy is prolonged to 630 cycles, twice as that with bare Zn. Supplementary Fig. 29 illustrates the typical CV curves of full cells, unveiling comparable energy storage behavior, with distinct vanadium-related redox peaks. The distinction lies in the fact that, compared to the bare Zn counterpart, Zn-Ti//$NH_4V_4O_{10}$ exhibits a diminished voltage gap between the cathodic and anodic peaks, coupled with an amplified current response. The improvement in reaction kinetics is also corroborated by impedance analysis, which demonstrates reduced charge transfer resistance and steeper slope in the relevant frequency regions (Supplementary Fig. 30)[56]. As a result, the rate evaluation of full cell with Zn-Ti alloy presents the slightly enhanced capacity retention. (Fig. 5a and Supplementary Fig. 31). In the case of long-term cycling, the Zn-Ti// $NH_4V_4O_{10}$ demonstrates outstanding capacity retention for nearly 3500 cycles, in accordance with the steady voltage hysteresis (Fig. 5b and Supplementary Fig. 32). This remarkable performance is attributed to the steady interfacial impedance, which should also translate to the compact and uniform morphology on the anode side (insets of Fig. 5b)[57]. However, the battery with bare Zn undergoes severe cell polarization after 700 cycles, followed by a rapid capacity degradation. Pits and protrusions were observed on the surface of bare Zn after cycling, stemming from the uncontrolled Zn plating/stripping

behavior. To validate the feasibility for scale-up, Zn-Ti alloy was incorporated into a multi-layer pouch cell. The corresponding battery configuration is presented in the inset of Fig. 5c, where multiple layers of $NH_4V_4O_{10}$ cathodes, separators and Zn-Ti alloy anodes are stacked together. By extending the mass loading of $NH_4V_4O_{10}$ to 6 mg cm$^{-2}$, a single-layer areal capacity of 1.42 mAh cm$^{-2}$ is obtained, culminating in a total discharge capacity of 34 mAh for the three-layer pouch cell (Fig. 5c). The voltage hysteresis remains substantially stable throughout the entire testing period, albeit with a slight increase relative to the coin cell (Supplementary Fig. 33). Consequently, a long-term cycling of over 500 cycles with 85% capacity retention is achieved, with only minor changes in voltage profiles. Such performance is well-positioned among state-of-the-art lab-level pouch cells reported in the literature (Fig. 5d and Supplementary Table 3). While further engineering efforts are required to construct high-energy-density AZBs with limited N/P ratios, lean electrolytes and large capacities that use the findings reported herein, our research provides a robust foundation for such efforts.

## Discussion

In this contribution, we presented an innovative GBE via alloying strategy to tailor the interfacial stability and deposition behavior of Zn anode. Multi-scale characterizations confirm the presence of Ti-containing IMCs in the as-fabricated Zn-Ti alloy, which preferentially accumulate at GBs. Comprehensive morphology analysis further reveals that IMCs could be thermodynamically stabilized in mildly acidic aqueous electrolytes independent of Zn migration, thereby greatly inhibiting the intergranular corrosion. Moreover, the well-defined surface structure modulates the initial Zn nucleation and growth behavior, as identified by classical crystallographic theory, leading to a hybrid 3DI + 3DP model. The partially progressive mode promotes the spatially uniform distribution of Zn nuclei, favoring a dense Zn deposition with low interfacial impedance. These merits enable highly reversible Zn plating/stripping for 4000 cycles with an average CE of 99.85%. Leveraging this high stability, AZBs with Zn-Ti alloy as opposed to conventional Zn anode exhibit twice the life span regardless of low or high rate. Promisingly, the multi-layer pouch cell retains 85% of its initial capacity after 500 cycles. Our findings open an

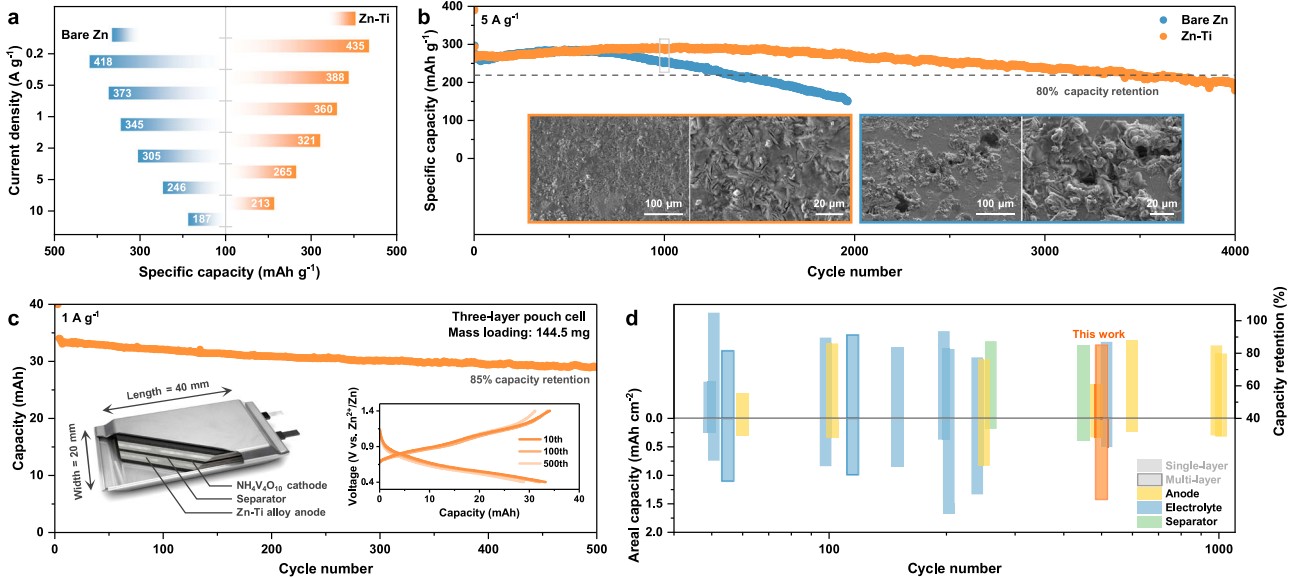

**Fig. 5 | Electrochemistry of Full Cells. a** Rate capability and **b** long-term cyclic stability (insets are SEM images of Zn anodes after 1000 cycles) of Zn//NH$_4$V$_4$O$_{10}$ coin-type full cells. **c** Cycling performance of the fabricated three-layer pouch cell (insets are corresponding voltage profiles and configuration diagram). **d** Comparison of cycling performance with other single/multi-layer pouch cells reported.

avenue for rational enhancement of the grain boundary stability in Zn anode, and could potentially be extended to other metal batteries using polycrystalline anodes.

# Methods

## Preparation of Zn-Ti alloy

Pure zinc and titanium were used as raw materials in a weight ratio of 99.5:0.5 to produce Zn-Ti alloy ingot through melting. The ingot was then wire-cut into sheets measuring 40×10×3 mm and polished until flat and smooth with fine sandpapers. Subsequently, these sheets were rolled to a thickness of 0.3 mm with a strain of ln(3/0.3) and annealed isothermally at 200 °C for 4 h. Finally, the resulting Zn-Ti alloy foils were cut into the required dimensions and polished for subsequent characterizations and electrochemical measurements.

## Synthesis of NH$_4$V$_4$O$_{10}$

NH$_4$V$_4$O$_{10}$ was synthesized based on a prior study[58]. Typically, 18 mM NH$_4$VO$_3$ and 27 mM H$_2$C$_2$O$_4$·2H$_2$O were dispersed in 90 mL deionized water at 80 °C. The mixture was then transferred to a Teflon-lined autoclave and heated at 140 °C for 48 h. The obtained product was collected by filtration, washed with deionized water and ethanol, and ultimately dried at 70 °C.

## Characterizations

The crystallographic phase and chemical composition were investigated by X-ray diffraction (XRD, Rigaku D/max2500) with Cu Kα radiation and inductively coupled plasma optical emission spectrometer (ICP-OES, Spectro Blue Sop). The surface topography, crystal orientation, microstructure and corresponding elemental mapping were characterized by field emission scanning electrode microscopy (SEM, TESCAN MIRA4 LMH) equipped with energy disperse spectroscopy (EDS, One Max 50), atomic force microscope (AFM, Bruker Dimension Icon), confocal laser scanning microscope (CLSM, KEYENCE VK-X1000), electron backscatter diffraction (EBSD, Nordlys-Max2), and transmission electron microscopy (TEM, FEI Tecani F30) equipped with an EDS detector (Oxford Xplore). Note that the EBSD and CLSM samples were subjected to conventional sanding and twin-jet electropolishing processes to obtain a smooth surface and diminish the extra stress. Operando visualization of Zn deposition behavoir was

carried out by coupling optical microscope (CEWEI LW750LJT) and custom-built visualization cell (BJSCISTAR).

## Electrochemical measurements

The Tafel, linear sweep voltammetry (LSV), cyclic voltammetry (CV) and chronoamperometry (CA) were conducted on an electrochemical workstation (Multi Autolab/M204) with a three-electrode system using Zn anode as working electrode, platinum as counter electrode and Ag/AgCl as reference electrode. The galvanostatic cycling in the form of coin cell (CR2025) was carried out using a testing system (LAND CT2001A) at room temperature of 25°C. The Zn//Zn symmetric cells were assembled with Zn foils (100 μm in thickness and 12 mm in diameter, unless otherwise specified) as both the working and counter electrodes, 3 M ZnSO$_4$ as the electrolyte, and glass microfiber filters (Whatman, GF/D) as the separator. For Zn//Cu asymmetric cells, Cu foil (10 μm in thickness) was employed as the working electrode, and the charge cutoff voltage was set to 0.5 V (vs. Zn$^{2+}$/Zn). The NH$_4$V$_4$O$_{10}$ cahode was fabricated with active materials, Ketjen Black, poly-vinylidene fluoride in a weight ratio of 7:2:1. The active mass loading of NH$_4$V$_4$O$_{10}$ was approximately 1 (for coin cell) or 6 mg cm$^{-2}$ (for pouch cell). The assembled Zn//NH$_4$V$_4$O$_{10}$ full cells were initially activated with one-tenth of the applied current prior to formal test. Subsequently, they were subjected to long-term cyclic stability testing within the voltage range of 0.4-1.4 V (vs. Zn$^{2+}$/Zn). The electrochemical impedance spectroscopy (EIS) was measured over a frequency range from 100 kHz to 10 mHz on a electrochemical workstation (CHI660E). The pouch cell (20×40 mm) was tested under specific pressure conditions using the multichannel battery tester (Neware CT-4008T) within a climatic chamber (BOLING BLC-300) matained at 25°C.

## Theoretical calculations

The density functional theory (DFT) calculations were conducted to investigate the adsorpotion behavoir of Zn atoms on Zn and TiZn$_{16}$ facets, using the Projector Augmented-Wave (PAW) pseudopotentials as implemented in the Vienna Ab-initio Simulation Package (VASP 5.4.4)[59,60]. The Generalized Gradient Approximation (GGA) as formulated by the Perdew-Burke-Ernzerhof (PBE) functional was employed to describe the exchange-correlation interaction[61]. A vacuum layer of 20 Å was incorporated along the z-direction to avoid

interactions from neighboring cells in a periodic boundary condition. The structures were relaxed until the total energy and residual force per atom converged to values below $1.0 \times 10^{-5}$ eV and $2.0 \times 10^{-2}$ eV Å$^{-1}$, respectively. An energy cutoff of 450 eV was applied for all calculations. The binding energy between the substrate and a Zn atom was defined as follows:

$$E_b = E_{sub-Zn} - E_{sub} - E_{Zn} \qquad (3)$$

where $E_{sub-Zn}$ refers the energy of Zn adsorbed substrate, $E_{sub}$ the energy of pritine substrate, and $E_{Zn}$ the energy of a single Zn atom.

### Finite element simulations

Three-dimensional (3D) models were established with COMSOL (COMSOL Multiphysics 5.6) electrochemistry module to understand the dynamics of Zn electrodeposition processes. Concretely, the electrochemistry was simulated using the Tertiary Current Distribution module (Nernst-Planck Equations interface), where the Deformed Geometry node kept track of the deformation of localized geometry structure during Zn deposition. Nernst-Planck equation was used to describe the diffusion and migration of ions in the electric field:

$$J_{Zn} = -D_{Zn}\nabla c_{Zn} - z_{Zn} u_{Zn} F c_{Zn} \nabla \phi_l \qquad (4)$$

where $J_{Zn}$ denotes the $Zn^{2+}$ flux, $D_{Zn}$ the diffusion coefficient, $c_{Zn}$ the electrolyte concentration, $z_{Zn}$ the charge number, $u_{Zn}$ the electric mobility, and $\phi_l$ the electrolyte potential. The boundary condition for Zn deposition was given by Butler-Volmer equation, and the local current density was calculated as follows:

$$i_{Zn} = i_0 \left[ \exp\left(\frac{\alpha_a F \eta}{RT}\right) - \exp\left(-\frac{\alpha_c F \eta}{RT}\right) \right] \qquad (5)$$

where $i_0$ represents the exchange current density, $\alpha$ the charge transfer coefficient, $\eta$ the overpotential, and $F/RT$ the Nernst parameter. The related physical parameters were defined and listed in Supplementary Table 4.

### Reporting summary

Further information on research design is available in the Nature Portfolio Reporting Summary linked to this article.

## Data availability

All data that support the findings of this study are presented in the Manuscript and Supplementary Information, or are available from the corresponding author upon reasonable request. Source data are provided with this paper.

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

## Acknowledgements

The research was financially supported by the National Natural Science Foundation of China (No. 51932011 (S.L.), 52202339 (X.Z.), and 52372252 (J.Z.)) and Hunan Natural Science Fund for Distinguished Young Scholar (No. 2021JJ10064 (J.Z.)). Meanwhile, the authors are grateful for computing resources from the High-Performance Computing Center of Central South University.

## Author contributions

X.Z. and J.Z. conceived the idea and designed the experiments. S.L. and J.Z. provided critical guidance on the project. Y.Z., X.Z. and M.C. carried out the synthesis, materials characterizations and electrochemical measurements. Y.Z. performed the finite element simulations. S.G. conducted the theoretical calculations. B.L. provided important feedback that facilitated a systematic data analysis. Y.Z. and X.Z. wrote the manuscript. All authors contributed to the discussion of the results.

## Competing interests

The authors declare no competing interests.
