## [Peer Review File · Nature Communications]

Tailoring grain boundary stability of zinc-titanium alloy for long-lasting aqueous zinc batteriesREVIEWER COMMENTS

Reviewer #1 (Remarks to the Author):

This manuscript, entitled “Tailoring grain boundary stability of Zn-Ti alloy for long-lasting aqueous Zinc batteries”, shows that the effective control technique for Zinc plating/stripping process through the formation of Zn-Ti alloy on the grain boundary. They describe the results obtained using this GBE technique and consider the modeling of nuclei formation and growth to be quite useful. However, The paper is too preliminary and too speculative in various aspects. Thus, the authors should address several concerns below and resubmit the manuscript before the consideration for publication in Nature Communications.

1. The authors have presented Figure 1b, which illustrates the formation of a Zn-Ti alloy along the grain boundary of zinc metal. However, it is crucial for the authors to provide additional evidence, including the EBSD image, XRD results, and electrochemical performance data of pure Zn metal produced using the same process. The XRD results of the Zn-Ti alloy demonstrate an increase in intensity specifically in the (002) plane, indicating a growing tendency of zinc metal towards this plane during the alloying process. Notably, several researchers have reported on the crystallographic strategy for zinc metal growth (ACS Nano 16, 9150–9162 (2022), Adv. Mater., 34, 2202552 (2022), eScience 100120 (2023); DOI: 10.1016/j.esci.2023.100120). Thus, it is necessary to provide an in-depth explanation concerning the growth orientation of zinc metal during the metallurgical procedure.
2. In Figure 2d, corrosion pits formed in the Zn-Ti alloy show a particularly concentrated and directional pattern. Authors need to investigate whether the corrosion resistance of Zn-Ti alloys is effectively preserved within this specific grain structure. Additionally, it would be helpful to provide information on the corrosion resistance of pure TiZn16 alloy in the ZnSO₄ electrolyte.
3. In Figure 3f, the formation of nuclei in a specific direction is depicted. It is important to interpret this observation and determine if there is a tendency towards a particular orientation resulting from the mechanical rolling process.
4. The figure has a low resolution, making it difficult for readers to read and understand. It is necessary to improve the resolution of the figure in order to enhance its readability.

Reviewer #2 (Remarks to the Author):

In this work, the authors reported a new way to increase the longevity of the aqueous zinc battery by tailoring grain boundary stability of Zn-Ti alloy. This idea is interesting and makes sense scientifically. The authors demonstrated the preferential distribution of intermetallic compounds at grain boundaries can substantially suppress intergranular corrosion. The manuscript is well prepared and presented. Readers can easily find useful information from the presented figures and supplementary. My questions mainly focus on the zinc battery modeling part.

- (1) It is not clear what is the criteria for the Zn deposition and growth? This is very important and can be understood from the thermodynamics perspective.
- (2) How did the authors determine the nucleation of the Zn? The nucleation model is nothing new there. More specifically, Eqs. (1-3) are to describe the number of nucleation. A more important is to know where the nucleation would occur.
- (3) It is challenging for me to correlate the model results with the experiment results. I

assume that the numerical model should provide insights for understanding the Zn deposition. However, I only see some superficial descriptions regarding the morphology on some time points and also confirm with the SEM figure. If that is the case, what is the point of employing the model? I expect a more thorough discussion from the numerical model to help improve the understanding (of course, we need to make sure that the model is correct).

(4) In Fig. 3 (d) and (e), how many cycles have been there? Have the authors consider the stripping process as well?

As such, I expect that the authors can explore more details from the numerical simulation to provide more knowledge in understanding the fundamental mechanism.

Reviewer #3 (Remarks to the Author):

This finding work demonstrated impressive results. The grain boundary engineering of Zn with Ti alloy are a simplified method but yield excellent results for Zn suppression. The well-designed experiment with the insight characterization reveals the novel finding in this field. Please check on my comments:

1. Why the author chose pure zinc and titanium in a weight ratio of 99.5:0.5 for this work. Is it an optimized condition? Is there any effect from grain size?
2. How the intensity peak of $Zn_4SO_4(OH)_6 \cdot H_2O$ in XRD pattern reflects the HER reaction, please explain and What state of sample is used to measure XRD?
3. In Fig. 2e, how to obtain the ratio of phase recognize? Does it represent only for interested area? Can we use operando XRD technique to confirm the bulk properties?
4. Could the author provide the corrosion rate from Tafel plot?
5. Why the CV curve in Supplementary Figure 9. was employed by using Ag/AgCl instead of Zinc reference.
6. It might be perfect to support the change of structure during cycling test and verify the by-product from parasitic reactions at Zn surface by ex-situ XPS along with XRD and SEM.
7. Favorable Nucleation and Growth Models used in this work can predict only at early state of Zn deposition or continue deposited Zn on substrate. Since Zn anode has deposition and dissolution during plating/stripping process, how about the explanation of the dissolution of Zn on Zn-Ti during stripping. Is there any benefit of Zn-Ti after Zn dissolution for the next cycle of plating process.
8. The value of y axis on supplementary Figure 11 might not be correct.
9. Fig. 3a, it is not clear between nucleation and growth.
10. Fig 3c, how to calculate Gib free energy? and also for $\Delta G_{nucleation}$ and ΔG_{growth} ? Please provide the information.
11. To compare the Zn growth on Zn-Ti in Fig. 3f, the magnification of SEM for Zn-Ti should be similar to Zn.
12. How such a Zn-Ti can reduce the significant voltage drop at the beginning of Zn deposition. Please provide more explanation.
13. Why the use of Zn-Ti anode can yield the high capacity and better rate performance than that of Zn anode? Please give an explanation.

Response to Reviewers' Comments

We would like to convey our sincere appreciation for the invaluable comments and insightful suggestions provided by the reviewers on our manuscript. We have taken full consideration of these comments, conducted additional experiments, and revised the manuscript accordingly. Please find below a comprehensive description of the changes that have been implemented, along with our point-by-point response to the reviewers' comments.

Review #1

This manuscript, entitled “Tailoring grain boundary stability of Zn-Ti alloy for long-lasting aqueous Zinc batteries”, shows that the effective control technique for Zinc plating/stripping process through the formation of Zn-Ti alloy on the grain boundary. They describe the results obtained using this GBE technique and consider the modeling of nuclei formation and growth to be quite useful. However, the paper is too preliminary and too speculative in various aspects. Thus, the authors should address several concerns below and resubmit the manuscript before the consideration for publication in Nature Communications.

Response:

We are glad to have the reviewer's acknowledgement on the major achievements of our study. Meanwhile, the critical feedback from the reviewer will contribute to further elucidating the findings and underlying mechanisms of this research, thereby enhancing the quality of our manuscript. Indeed, this study focuses on two primary issues. Firstly, the inherent polycrystalline nature of commercial Zn anodes drives the corrosion reactions inevitably affected by the surface microstructure, with grain boundaries, possessing high susceptibility to corrosion (*Journal of the American Chemical Society, 2021, 143, 3143-3152*), being particularly affected. Secondly, scant attention has been directed towards the initial nucleation and growth behavior, neglecting a critical scientific foundation of Zn deposition. In light of this, we proposed an innovative grain boundary engineering approach via alloying strategy, to simultaneously tailor the interfacial stability and promote the favorable Zn deposition, which has been demonstrated to enable long-lasting aqueous Zn batteries. In response to

the reviewer's valuable and insightful suggestions, we have diligently revised the manuscript and included additional evidence crucial to the integrity of this study, with substantial revisions made in the "Corrosion Resistance via Grain Boundary Engineering" and "Favorable Nucleation and Growth Models" sections. By clarifying these issues that were not adequately resolved in the previous version, we sincerely hope that you will reconsider the publication of this work in *Nature Communications*.

1. The authors have presented Figure 1b, which illustrates the formation of a Zn-Ti alloy along the grain boundary of zinc metal. However, it is crucial for the authors to provide additional evidence, including the EBSD image, XRD results, and electrochemical performance data of pure Zn metal produced using the same process. The XRD results of the Zn-Ti alloy demonstrate an increase in intensity specifically in the (002) plane, indicating a growing tendency of zinc metal towards this plane during the alloying process. Notably, several researchers have reported on the crystallographic strategy for zinc metal growth (ACS Nano 16, 9150–9162 (2022), Adv. Mater., 34, 2202552 (2022), eScience 100120 (2023); DOI: 10.1016/j.esci.2023.100120). Thus, it is necessary to provide an in-depth explanation concerning the growth orientation of zinc metal during the metallurgical procedure.

Response:

We appreciate the reviewer for mentioning certain issues concerning the influence of metallurgical treatments on the texture of Zn metal anodes, particularly highlighting the (002) crystallographic plane. Indeed, the increase in the intensity of (002) diffraction peak originates from the rolling procedure employed in alloy processing, as described in the experimental section. Orientational ordering phenomenon occurs, when anisotropic material undergoes plastic deformation under an externally applied stress. Among various plastic deformation techniques such as rolling, extrusion, and forging, rolling appears to be most relevant to the manufacture of textured metal electrode materials (*Chemical Reviews*, 2022, 122, 14440-14470). The texture evolution induced by plastic deformation largely depends on the deformation modes (slip and twinning) and intrinsic crystal symmetry (**Fig. R1a**). In the most conventional scenario, slip occurs preferentially at

close-packed planes along the close-packed directions, such as the (002) plane along the $\langle 110 \rangle$ direction in hexagonal close-packed (HCP) metals (**Fig. R1b**) (*Advanced Materials*, 2022, 34, 2106867). This slip mechanism in Zn metal results in a characteristic (002) texture (*ACS Energy Letters*, 2022, 7, 3564-3571). In our previous study (*the results to be published*), as depicted in the pole figures (**Fig. R1c**), the texture intensity for (002) plane of bare Zn increased from 5.86 to 18.71 after rolling, confirming that rolling procedure affects the crystallographic orientation of Zn metal anodes. According to the research by Archer et al., even with 1 time of rolling, there is a noticeable enhancement in the intensity of (002) diffraction peak, and with repeated rolling (severe plastic deformation), nearly single (002)-textured Zn anode can be achieved (**Fig. R1d**). However, the metallurgical treatment in this study does not involve severe plastic deformation. The rolling procedure (1 time) conducted here aims to obtain Zn foils from ingots for battery testing, which offers a path to precisely control the thickness of metal electrode. Similar processing-induced texture changes were also observed in lithium alloys, with the distinction being that due to the body-centered cubic lattice symmetry of lithium, a stronger (200) texture was exhibited (*Nature Communications*, 2020, 11, 829; *Journal of the American Chemical Society*, 2020, 142, 2438-2447).

Figure R1. (a) Dominant deformation mechanisms in HCP metals. (b) Slip mechanism in Zn metal

and schematic diagram of the rolling procedure. (c) Pole figures of bare Zn before and after rolling.

(d) XRD patterns of Zn foil subjected to repeated rolling.

In general, (002) texture is believed to be responsible for the favorable Zn deposition through epitaxial growth (*Science Bulletin*, 2022, 67, 716-724). Nevertheless, in our study, despite the initial texture differences between bare Zn and Zn-Ti alloy, with the (002) plane slightly elevated in the latter, no significant texture changes were observed either after 1 h of deposition or after 100 h of cycling in the symmetric cells (**Fig. R2a**). We strongly concur that a single (002)-textured Zn anode can significantly enhance the cycling stability of Zn batteries. However, we maintain a cautious stance regarding the substantial benefits brought about by the slight increase in (002) texture. In fact, this might be a viewpoint that has been misunderstood all along. One of the references mentioned by the reviewer provides a detailed elucidation of this issue (*Advanced Materials*, 2022, 34, 2202552). In the case of the semi-matched substrate, such as polycrystalline Zn, regardless of whether the (002) diffraction peak intensity is high or low, imperfect interfaces are formed due to the lattice mismatch (**Fig. R2b**) When two Zn islands encounter, defects arise, eventually leading to irregular Zn deposition. Once such non-planar deposits form, they disrupt the initially homogenous surface, weakening the preference for uniform planar electrodeposition in subsequent cycles, thereby constraining the electrode's ability for repetitive cycling. Our latest research has unveiled the underlying mechanism, revealing that only single (002)-textured Zn anode sustain its effectiveness (*Nature Portfolio, In Revision*). In situations involving even minor lattice mismatch, once the overgrowth crystal surpasses the critical thickness t_c , the subsequently deposited Zn tends to become disordered and irregular. This phenomenon aligns with the Royer's theory. The t_c for the overgrowth crystal can be correlated to the lattice mismatch by (*Physica Status Solidi B*, 1967, 19, 95-105; *International Journal of Materials Research*, 2022, 92, 701-706):

$$t_c = b \left(\frac{f_s}{f} \right)^2 \quad (1)$$

where f depends only on the lattice parameters (a , b) of the substrate and the epitaxial layer ($f = (b - a)/a$). As demonstrated by **Fig. R3**, even with a strong (002)-textured Zn (rather than single (002)-textured), pronounced non-planar deposits can be observed during the deposition process. This incomplete texture within the Zn anode could be “overwritten” by repeated plating/stripping upon

prolonged cycling, ultimately leading to electrode failure. Based on the above analysis, we intuitively conclude that the impact of a single pass of rolling on the texture of Zn-Ti alloy, and subsequently on the electrochemical performance of the battery could be deemed negligible, without compromising the core findings and conclusions of this study.

Figure R2. (a) XRD patterns of bare Zn and Zn-Ti alloy at pristine state, after deposition for 60 min and after cycling for 100 h. (b) Schematic diagram of Zn electrodeposition onto a semi-matched substrate.

Figure R3. (a) XRD patterns and SEM images of the single (002)-textured Zn anode (b) before and (c) after deposition for 30 min. (d) XRD patterns and SEM images of the strong (002)-textured Zn anode (e) before and (f) after deposition for 30 min.

More importantly, we would like to emphasize the innovation of this study and reiterate that the significant enhancement in the stability of Zn batteries is attributed to the grain boundary engineering. Multi-scale characterization techniques (EBSD, BSE/SE-SEM, TEM) demonstrated the preferential distribution of TiZn_{16} intermetallic compounds along the grain boundaries. Electrochemical tests (Tafel, LSV), immersion experiments, and post-mortem analysis of cycled electrode validated the substantial suppression of HER-induced corrosion reactions. Remarkably, with the aid of EBSD and CLSM, we elucidated that the corrosion reactions in polycrystalline Zn primarily occur at the grain boundaries. This Zn-Ti alloy with a specific grain boundary structure fundamentally inhibited the intergranular corrosion and mitigated the loss of active Zn. Furthermore, time-series SEM images and corresponding statistical results revealed the disparities in morphological evolution during deposition processes, stemming from the reduced energy difference between Gibbs free energy for Zn nucleation and growth, corroborated by characteristic overpotential analysis and DFT calculations. Concurrently, a hybrid 3DI+3DP nucleation mode was unveiled through chronoamperometry, and the superiority in subsequent growth process was further affirmed by COMSOL simulations. These merits enable highly reversible Zn plating/stripping for 4000 cycles with an average CE of 99.85% at

5 mA cm⁻², corresponding to a total cumulative plated capacity of 4 Ah cm⁻². The fabricated multi-layer pouch cell maintains 85% of its initial capacity after 500 cycles.

2. In Figure 2d, corrosion pits formed in the Zn-Ti alloy show a particularly concentrated and directional pattern. Authors need to investigate whether the corrosion resistance of Zn-Ti alloys is effectively preserved within this specific grain structure. Additionally, it would be helpful to provide information on the corrosion resistance of pure TiZn₁₆ alloy in the ZnSO₄ electrolyte.

Response:

Thank you for the reviewer's reminder. We apologize for any misunderstanding that may have arisen, due to our inadequate description and potentially unclear images. In this immersion experiment, to obtain high-quality EBSD IPF mapping images and accurately identify the corrosion initiation sites, corrosion products on the surface of Zn anodes were removed. It should be noted that EBSD analysis faces challenges in resolving the rough regions and the precipitated second phase, and the samples have been subjected to twin-jet electropolishing prior to immersion. Therefore, the unrecognizable white regions observed on the Zn anodes after immersion can be attributed to either TiZn₁₆ intermetallic compounds or corrosion pits (**Fig. 2f**). Additional SEM images were employed to differentiate between them, with the former showing strip-like features continuously distributed along grain boundaries, while the latter typically displayed irregular shapes (**Supplementary Fig. 11**). Evidently, the Zn-Ti alloy demonstrates only minor signs of corrosion, indicating the effective suppression of corrosion by TiZn₁₆ at the grain boundaries. In contrast, the surface of bare Zn shows corrosion pits clustered along the grain boundaries, confirming severe intergranular corrosion, which aligns with the EBSD observations. Based on these analyses, we have reasonably identified the unrecognizable white parts in **Fig. 2f**, as indicated by the arrows and labels. In addition, similar results were also corroborated via CLSM, where discernible corrosion pits were observed on the surface of bare Zn, in stark contrast to the relatively smooth appearance of Zn-Ti alloy (**Fig. 2h-i**).

According to the Zn-Ti binary phase diagram (**Supplementary Fig. 2**) and the microstructural analysis shown in **Fig. 1**, the prepared Zn-Ti alloy consists of the Zn-rich solid solution matrix (with negligible titanium content) and the second phase of TiZn₁₆ intermetallic compounds. Typically, the

matrix and the second phase are closely combined, making separation challenging. One potential approach is to exploit the distinct corrosion rates of the matrix and second phase in an acidic solution (*Journal of Solid State Chemistry*, 1995, 118, 219-226). Upon subjecting the Zn-Ti alloy to twin-jet electropolishing, employing a perchloric acid ethanol solution as the electrolyte, a significant height contrast between the second phase and the matrix becomes evident (**Fig. R4**). This observation highlights that the TiZn_{16} intermetallic compounds are less susceptible to corrosion compared to the Zn matrix, thus confirming their superior corrosion inhibition capability. In addition, the diffraction peaks corresponding to TiZn_{16} intermetallic compounds retain their intensity regardless of aging or extended cycling, providing additional evidence of their robust corrosion resistance (**Fig. 2b** and **Supplementary Fig. 9**).

Figure R4. (a) BSE-SEM and (b) SE-SEM images of Zn-Ti alloy after twin-jet electropolishing.

Additionally, we have supplemented ex-situ XRD, SEM, and optical images of the Zn anodes immersed in ZnSO_4 electrolytes for different durations, to further confirm the excellent corrosion inhibition ability of Zn-Ti alloy. It should be noted that, unlike the results shown in **Fig. 2f** and **Supplementary Fig. 11**, where the surface by-products were removed to investigate the origin of corrosion, the present experiments aim to preserve the original corrosion morphology to the greatest extent possible. As illustrated in **Fig. 2b** and **Supplementary Fig. 8**, even after immersing the Zn-Ti alloy in ZnSO_4 electrolyte for 10 days, only minimal by-products can be observed on its surface, along with relatively weak diffraction peaks of zinc hydroxysulfate. These findings once again validate the effectiveness of Zn-Ti alloy in inhibiting the corrosion reaction.

Changes:

To further clarify the significance of GBE in inhibiting intergranular corrosion, EBSD analysis was performed to monitor the reaction process, in which the phase recognition was quantified on the basis of image identification. As shown in **Fig. 2f**, the scanning pattern with color mapping is identified as active Zn, whereas the white parts correspond to the unrecognizable phase that can be attributed to TiZn_{16} IMCs or corrosion pits. To discern between these phases, additional SEM images were employed, with the former revealing a continuous distribution (**Supplementary Fig. 11**). The EBSD IPF mapping of both bare Zn and Zn-Ti alloy is clear at pristine state. Afterwards, the bare Zn surface displays a typical localized corrosion morphology. The most pronounced corrosion damage manifests along the GBs, followed by its expansion towards the interior of grains or even deeper regions, as indicated by arrows. In contrast, minimal signs of corrosion are observed on Zn-Ti alloy, as further confirmed by the ratio of phase recognition, which only marginally decreases from 98% to 95% after immersion (**Fig. 2g**). However, the recognizable phase of bare Zn decreases sharply from nearly 100% to 89%. The distinct corrosion morphology was then verified through confocal laser scanning microscope (CLSM) imaging, where irregular pits with depths ranging from 1 to 2 μm are discernible on the surface of bare Zn, signifying the initiation of micro-pitting corrosion (**Fig. 2h**). Conversely, owing to the suppressed localized corrosion, Zn-Ti alloy retains a relatively flat appearance with low roughness (**Fig. 2i**). (In the Manuscript, Page 8)

To accurately identify the corrosion initiation sites and obtain high-quality SEM and EBSD IPF mapping images, corrosion products on the surface of Zn anodes were removed with deionized water. Additional SEM images were employed to differentiate unrecognized white parts within the EBSD images and to provide supplementary information about the corrosion behavior. Specifically, the unrecognized phase, characterized by its continuous distribution along GBs, can be attribute to TiZn_{16} IMCs, while the irregularly shaped ones are identified as corrosion pits. (In the Supporting Information, Page R9)

Fig. 2 | Suppression of Parasitic Reaction through Grain Boundary Engineering. **a** Tafel plots. **b** Ex-situ XRD patterns of Zn anodes after immersion in ZnSO₄ electrolyte for different days. **c** Cross-sectional SEM images of Zn anodes after 50 cycles. Schematic diagram of **d** general corrosion and **e** intergranular corrosion. **f** EBSD IPF mapping images and corresponding **g** ratio of phase recognition of Zn anodes at pristine state and after 24 h of immersion. CLSM imaging of corrosion morphologies on **h** bare Zn and **i** Zn-Ti alloy, and corresponding surface profiles extracted along the location marked. **j** “Reservoir” protocol for evaluating average Zn plating/stripping CE.

The propensity of the Zn anode to degrade under prolonged aging and cycling in mildly acidic electrolyte, owing to reactions with water solvent, results in the irreversible consumption of metallic Zn into by-products³⁹. To evaluate the shelf life, Zn anodes were immersed in ZnSO₄ electrolytes for varying durations. It is evident that for bare Zn, the diffraction peaks corresponding to Zn₄SO₄(OH)₆·3H₂O (PDF#39-0689) begin to appear after 1 day of immersion and become significant after 10 days, indicating severe corrosion reactions (Fig. 2b). The corresponding SEM

and EDS mapping further corroborate this result, revealing discernible flake-like by-products and noticeable signs of corrosion (Supplementary Fig. 8). Conversely, the surface of Zn-Ti alloy displays minimal by-products, with the diffraction peaks of zinc hydroxysulfate remaining inconspicuous even after 10 days of immersion, demonstrating the excellent corrosion inhibition capability. (In the Manuscript, Page 7)

Supplementary Figure 8. (a) Optical photograph of the Zn anodes during immersion. SEM and corresponding EDS elemental mapping of (b-c) Zn-Ti alloy and (d-e) bare Zn after 10 days of immersion.

It is noteworthy that during the immersion process, the surface of Zn-Ti alloy exhibits only a few sporadic bubbles, in sharp contrast to the evident bubble accumulation observed on bare Zn. (In the Supporting Information, Page R6)

3. In Figure 3f, the formation of nuclei in a specific direction is depicted. It is important to interpret this observation and determine if there is a tendency towards a particular orientation resulting from the mechanical rolling process.

Response:

We express our gratitude to the reviewer for thoroughly examining our manuscript and for

presenting the insightful inquiries. In reality, the observed inclination toward "formation of nuclei in a specific direction" originates from partial nucleation occurring along the scratches, a phenomenon well-documented in numerous literature sources (*Nature Communications*, 2023, 14, 76; *Nature Communications*, 2020, 11, 1634; *Energy & Environmental Science*, 2022, 15, 1638-1646; *Angewandte Chemie*, 2022, 61, e202212587). Regrettably, the generation of these scratches is unavoidable, as the emulsion oil used during the wire cutting process can contaminate the alloy surface. As a result, it is necessary to polish the samples with low-grit 2000-mesh sandpaper to eliminate the surface impurities, as described in the experimental section. What we would like to convey is that, the scratches seem to have a limited impact on the nucleation and growth modes in this study. As depicted in **Fig. R5**, the initial nucleation behavior of both polished and unpolished bare Zn is strikingly similar, featuring limited nucleation sites. Upon extending the electrodeposition time to 60 min, the surface displays an uneven morphology with distinct protrusions and numerous uncovered areas.

Figure R5. SEM images of (a-c) unpolished bare Zn and (d-i) polished bare Zn after deposition for 15 s, 30 s and 60 min.

Moreover, we wish to reiterate the underlying factors that account for the contrasting nucleation and growth mechanisms observed in the two Zn anodes, factors that are independent of the processing procedure. Considering the inadequate descriptions and insufficient discussions in the initial version, we have restructured the "Favorable Nucleation and Growth Models" section and have included additional details that are critical to the integrity of this section. In this section, we firstly examined the morphological evolution during the nucleation and growth processes, as illustrated by the time-series SEM images (**Fig. 3a-b** and **Supplementary Fig. 12-13**). For bare Zn, Zn nuclei randomly appear at the limited nucleation sites, leaving behind widely uncovered regions. Subsequently, Zn^{2+} exhibits a preference to deposit onto existing nuclei, leading to the pronounced protrusions. While for Zn-Ti alloy, as the deposition moves forward, the number and size of Zn nuclei increase simultaneously. The abundant Zn nuclei then navigate the ensuing growth process. As neighboring particles come into contact with each other, the deposited Zn particles gradually merge together and cover the whole surface of substrate. To elucidate the observed disparities in deposition morphology, we analyzed the voltage profiles obtained from galvanostatic electrodeposition (**Supplementary Fig. 15**). The Zn-Ti alloy demonstrates a small nucleation overpotential, suggesting a lower energy barrier. The distinctions in nucleation overpotentials can be attributed to zincophilicity, whereby $TiZn_{16}$, possessing a more negative binding energy, could potentially serve as nucleation sites to facilitate the process (**Supplementary Fig. 16**). More importantly, the much-reduced overpotential difference (gap between nucleation and plateau overpotentials) signifies that, for Zn-Ti alloy, nucleation is more prone to occur compared to bare Zn (**Fig. 3d**). Subsequently, we employed chronoamperometry to investigate the initial nucleation model (**Fig. 3e** and **Supplementary Fig. 17**). The normalized dimensionless transient for bare Zn is well-fitted by the 3DI model, while Zn-Ti alloy follows a hybrid 3DI+3DP model. The 3DI mode implies that nuclei appear almost instantaneously at all possible but limited sites and then grow, whereas the partially progressive mode signifies nucleation sites being incrementally activated, endowing the Zn-Ti alloy with abundant Zn nuclei. These model results align well with the observed morphological disparities

during the initial deposition process (**Fig. 3a**). Evidently, the nucleation modes will inevitably influence the local electrochemical environment, and thus the subsequent Zn deposition. The growth process following nucleation was simulated using COMSOL (**Fig. 3f-g** and **Supplementary Fig. 18-19**). The Zn nuclei formed at the early stage will interfere with the electric field and ion flux distribution at the interface. Since the priority of Zn growth depends on this distribution, the abundant and evenly distributed Zn nuclei on Zn-Ti alloy can diminish the heterogeneity of further growth, thereby developing into a relatively flat deposited layer. These simulated results are in accordance with the morphological evolution depicted in **Fig. 3b**. Based on the above observations and analyses, the nucleation and growth mechanisms were deduced schematically in **Fig. 3h**, where solid circles with centers represent "nucleation" and dashed circles symbolize "growth".

4. The figure has a low resolution, making it difficult for readers to read and understand. It is necessary to improve the resolution of the figure in order to enhance its readability.

Response:

Thanks for your constructive comment. We apologize for the inconvenience caused by our inadequate image processing, and have taken necessary steps to improve the resolution of the figures in the revised version. Once again, we genuinely appreciate your understanding and the opportunity you provided us to enhance the quality of our work.

Review #2

In this work, the authors reported a new way to increase the longevity of the aqueous zinc battery by tailoring grain boundary stability of Zn-Ti alloy. This idea is interesting and makes sense scientifically. The authors demonstrated the preferential distribution of intermetallic compounds at grain boundaries can substantially suppress intergranular corrosion. The manuscript is well prepared and presented. Readers can easily find useful information from the presented figures and supplementary. My questions mainly focus on the zinc battery modeling part.

Response:

Thanks a lot, to the reviewer for acknowledging the novelty and scientific merit of this work, which focuses on tailoring grain boundary stability to achieve highly reversible Zn anodes. We genuinely value the opportunity for revision provided by the reviewer, and have included more evidence that is critical to the integrity of this study. In particular, following your kind suggestions, we have thoroughly revised the zinc battery modeling part to enhance the readability of this manuscript.

1. It is not clear what is the criteria for the Zn deposition and growth? This is very important and can be understood from the thermodynamics perspective.

Response:

Thanks for the reviewer's constructive comment. Typically, the Zn deposition in rechargeable cells initiates through heterogeneous nucleation on the surface of Zn metal anode substrates. The characteristics of substrate, such as composition, morphology, and surface chemistry, all have direct impacts on the distribution of concentration/electric fields around the substrate, thus influencing the Zn nucleation and subsequent growth process (*JACS Au*, 2023, 10.1021/jacsau.3c00292). Thermodynamically speaking, nucleation is determined by the decrease in bulk free energy (both chemical and electrical) and the increase in interfacial energy due to the formation of new interface (*Chemical Reviews*, 2021, 121, 5986-6056). The former serves as the driving force, while the latter constitutes the origin of nucleation barrier. According to classical nucleation theory, in the case of heterogeneous nucleation, the critical nucleation energy is derived as (*Science Advances*, 2019, 5,

eaau7728):

$$\Delta G_{\text{het}}^* = \frac{(2 - 3 \cos \theta + \cos^3 \theta)}{4} \Delta G_{\text{hom}}^* \quad (1)$$

where θ is the contact angle between Zn nucleus and the substrate (**Fig. R1**). Considering the quantitative relationship between interfacial tension and contact angle, expressed as:

$$\cos \theta = \frac{\gamma_{\text{SE}} - \gamma_{\text{SN}}}{\gamma_{\text{NE}}} \quad (2)$$

When the binding energy between Zn nucleus and the substrate becomes more negative, indicating a zincophilic nature, γ_{NE} is reduced and thus $\cos \theta$ is increased. Consequently, the nucleation energy barrier is decreased, leading to nucleation occurring at a lower nucleation overpotential. Therefore, a strong Zn binding ability, namely zincophilicity, is considered thermodynamically favorable for nucleation. We then conducted additional calculations to ascertain the binding energies between Zn atom and Zn/TiZn₁₆ facets. DFT calculations reveal that, compared to the Zn facets, the TiZn₁₆ facets exhibit more negative binding energies, endowing the Zn-Ti alloy with higher zincophilicity and thus a favorable Zn deposition (**Supplementary Fig. 16**).

Figure R1. Schematic diagram of Zn heterogeneous nucleation on the substrate.

On the other hand, the quality of initial nucleation plays an important role in the subsequent growth process. Finite element simulation results indicate that the newly formed Zn nuclei strongly interfere with the electric field and ion flux distribution at the interface (*Angewandte Chemie*, 2020, 59, 13180-13191). Since the priority of Zn growth relies on this distribution, a favorable nucleation mode can minimize the inhomogeneity of further growth, thereby yielding the relatively flat Zn deposition layer. As a consequence, in the following sections of this manuscript, we delve into the potential nucleation models through chronoamperometry (**Fig. 3e** and **Supplementary Fig. 17**) and simulate the post-nucleation growth process using COMSOL Multiphysics (**Fig. 3f-g** and

Supplementary Fig. 18-19) to validate this hypothesis. Undoubtedly, the abundant and evenly distributed Zn nuclei are advantageous for the desired Zn deposition.

Changes:

The nucleation of Zn nuclei involves a highly intricate process. From a thermodynamic perspective, the stability of a Zn nucleus deposited on the electrically charged substrate is determined by the combination of bulk free energy, comprising both chemical and electrical, and the interfacial free energy arising from the generation of new interfaces, which can be expressed by⁹:

$$\Delta G_{\text{sys}} = \left(\Delta G_V + \frac{zF\eta}{V_m} \right) S_V r^3 + (\gamma_{\text{NE}} S_{\text{NE}} + \gamma_{\text{SN}} S_{\text{SN}} - \gamma_{\text{SE}} S_{\text{SN}}) r^2 \quad (2)$$

$$S_V = \frac{\pi}{3} (2 - 3 \cos \theta + \cos^3 \theta) \quad (3)$$

$$S_{\text{NE}} = 2\pi(1 - \cos \theta) \quad (4)$$

$$S_{\text{SN}} = \pi(1 - \cos^2 \theta) \quad (5)$$

where ΔG_V stands for the volume free energy change (from liquid phase to solid phase), z the number of electrons transferred, F the Faraday's constant, η the overpotential, V_m the molar volume of Zn, S_V the volume factor of spherical cap, S_{NE} the curved surface area factor, S_{SN} the bottom surface area factor, γ_{NE} the nucleus/electrolyte interfacial free energy, γ_{SN} the substrate/nucleus interfacial free energy, γ_{SE} the substrate/electrolyte interfacial free energy, θ the contact angle, and r the radius of curvature of the nucleus. By incorporating the quantitative relationship between interfacial tension and contact angle, known as the Young's equation:

$$\gamma_{\text{SN}} - \gamma_{\text{SE}} = -\gamma_{\text{NE}} \cos \theta \quad (6)$$

Equation (2) can be expressed as follows:

$$\Delta G_{\text{sys}} = \left(\Delta G_V + \frac{zF\eta}{V_m} \right) S_V r^3 + \gamma_{\text{NE}} (S_{\text{NE}} - S_{\text{SN}} \cos \theta) r^2 \quad (7)$$

Physically, nucleation is thermodynamically favored when the volumetric and interfacial tension contributions to the energetics of system overcome the critical free energy. Given $d\Delta G_{\text{sys}}/dr = 0$, the critical radius required to form a thermodynamically stable atomic cluster can be derived as:

$$r_{\text{crit}} = \frac{-2\gamma_{\text{NE}} V_m}{zF\eta + \Delta G_V V_m} \quad (8)$$

This indicates that, for the same volume free energy change, the driving force for nucleation is dominated by the bulk electrostatic energy. Here, η_0 is defined as the characteristic overpotential:

$$\eta_0 = \frac{\Delta G_V V_m}{zF} \quad (9)$$

and therefore, the dimensionless overpotential can be written as:

$$\hat{\eta} = \frac{\eta}{\eta_0} \quad (10)$$

By substituting Equation (8) into Equation (7), the critical free energy to heterogeneously form a nucleus is given by:

$$\Delta G_{\text{crit}} = \frac{16\pi\gamma_{\text{NE}}^3 (2 - 3 \cos \theta + \cos^3 \theta)}{3\Delta G_V^2 4(1 + \hat{\eta})^2} \quad (11)$$

and expressed in dimensionless form:

$$\Delta \hat{G}_{\text{crit}} = \frac{\Delta G_{\text{crit}}}{\Delta G_0} = \frac{(2 - 3 \cos \theta + \cos^3 \theta)}{4(1 + \hat{\eta})^2} \quad (12)$$

where ΔG_0 is the characteristic Gibbs free energy of the system, which also corresponds to the critical Gibbs free energy for homogeneous nucleation in the electrolyte ($\theta = 180^\circ$) at $\eta = 0$. Calculations demonstrate that, for the system with a small contact angle (internal cause), namely zincophilic substrate, the critical free energy is reduced and nucleation occurs at the overpotential of smaller magnitude. (In the Supporting Information, Page R13)

The significant differences in the nucleation overpotentials are thought to be associated with the distinct affinities toward Zn⁴⁷. To gain a deeper insight into zincophilic behavior, the binding energies of Zn atom attached to the substrates were evaluated using density functional theory (DFT) calculations, with the selection of crystal planes guided by XRD results. As summarized in **Supplementary Fig. 16**, the calculated binding energy values between Zn atom and TiZn₁₆ (241) facet, (134) facet, and (313) facet are -1.07, -1.71, and -1.20 eV, respectively, which are more negative than those between Zn atom and Zn facets (-0.26, -0.66 and -0.85 eV). This indicates that TiZn₁₆ IMCs could potentially serve as nucleation sites, which endow the Zn-Ti alloy with high zincophilicity to facilitate Zn deposition, thus leading to a reduced nucleation overpotential and a lower energy barrier. (In the Manuscript, Page 11)

Supplementary Figure 16. (a) Optimal models of Zn atom adsorption on Zn (002) facet, Zn (100) facet, Zn (101) facet, TiZn₁₆ (241) facet, TiZn₁₆ (134) facet and TiZn₁₆ (313) facet. (b) Calculated binding energies of Zn atom with different crystal facets.

The selection of crystal facets is based on the three most prominent diffraction peaks identified in the XRD results. In the schematic diagram of the optimized model, Zn and Ti atoms are colored in gray and blue, respectively, while the adsorbed Zn atom is highlighted in orange. (In the Supporting Information, Page R16)

2. How did the authors determine the nucleation of the Zn? The nucleation model is nothing new there. More specifically, Eqs. (1-3) are to describe the number of nucleation. A more important is to know where the nucleation would occur.

Response:

Thanks for your valuable suggestions. In a typical chronoamperometric test, the working electrode is held below the equilibrium potential of Zn deposition (**Supplementary Fig. 17**). The current initially rises to a peak, which corresponds to an increase in the electroactive area associated with the growth of existing nuclei and generation of new nuclei (*Electrochimica Acta*, 1983, 28, 879-889). Subsequently, the current falls and approaches a steady value, suggesting a diffusion-controlled growth process. Obviously, nucleation and growth are two independent yet interrelated processes, and in practical scenarios, their boundaries become blurred. In our manuscript, the time used to distinguish between nucleation and growth corresponds to the descending part of the transients, precisely the moment just beyond t_m , when nucleation is reasonably well established and the Zn nuclei attain thermodynamic stability. (*Journal of The Electrochemical Society*, 1998, 145,

4090; *Energy & Environmental Science*, 2021, 14, 4077-4084). Similarly, in a galvanostatic electrodeposition test, nucleation is regarded as the initial voltage drop (**Supplementary Fig. 15**). Based on the above analysis, we have chosen two time points, 15 s and 30 s, to represent the morphology of nucleation (**Fig. 3a**).

Equation (1) was employed to depict the relationship between nuclei density and deposition time, aiming to differentiate between instantaneous nucleation and progressive nucleation. Instantaneous nucleation refers to the abrupt appearance of a limited number of nuclei right after the initial voltage drop, whereas progressive nucleation involves a gradual increase in the number of nuclei over an extended period. As this portion is widely recognized, it has been omitted from the manuscript. Furthermore, to quantitatively analyze the evolution of nuclei during the nucleation process, we have employed StarDist, a plugin for ImageJ, to assess the changes in both size and quantity of Zn nuclei over time. The histograms reveal that for Zn-Ti alloy, the number and size of Zn nuclei increase over time, indicative of a typical progressive feature (**Supplementary Fig. 13**). In contrast, for bare Zn, the emphasis primarily lies in the growth in size of existing nuclei.

To comprehensively elucidate the discrepancies observed in the initial deposition morphologies, current-time transients were normalized using the parameters j_m and t_m , enabling an investigation into nucleation modes. Building upon the theoretical framework proposed by Scharifker and Hills, Equations (2-3) delineate two extreme scenarios of nucleation, namely instantaneous nucleation and progressive nucleation. By comparing the experimental dimensionless transients with the theoretical models, we can gain insights into the potential nucleation mechanisms (**Fig. 3e**). Furthermore, it is essential to underscore the necessity of investigating the quantity of Zn nuclei, given that the initial nucleation can exert a significant influence on the subsequent growth process, as confirmed by the finite element simulation results (**Fig.3f-g** and **Supplementary Fig. 18-19**). Benefitting from the hybrid 3DI+3DP nucleation mode, the abundant Zn nuclei present on Zn-Ti alloy can interact with electric field and ion flux at the interface, mitigating the inhomogeneity of further growth. In contrast, the limited nucleation sites on bare Zn induce the localized deposition, that is, Zn^{2+} deposits onto the sparse Zn nuclei along the trajectory defined by electrolyte current density vector, resulting in pronounced protrusions.

In general, Zn^{2+} are thermodynamically favored to deposit at zincophilic sites due to the lower nucleation energy barrier (*JACS Au*, 2023, 10.1021/jacsau.3c00292; *Advanced Materials*, 2021, 33, 2004128). As per your kind suggestion, we have conducted additional calculations of the binding energies between Zn atom and Zn/TiZn₁₆ facets to explore the potential nucleation sites. DFT calculations reveal that, in comparison to Zn facets, TiZn₁₆ facets exhibit more negative binding energies, implying their potential to offer additional nucleation sites for promoting heterogeneous nucleation (**Supplementary Fig. 16**).

Changes:

All the transients exhibit a peak current (j_m), indicating the typical three-dimensional (3D) nucleation process with hemispherical diffusion control (**Supplementary Fig. 17**)⁵⁰. Initially, the rise in current reflects the expansion of electroactive area, owing to the growth in size of existing nuclei and the generation of new nuclei on the surface⁵¹. Subsequently, the current decreases and approaches a steady state due to the overlapping of neighboring nuclei and their diffusion zones, indicating a diffusion-controlled growth process⁵². It is evident that the Zn-Ti alloy demonstrates a slower and more progressive nucleation process in comparison to bare Zn, as substantiated by the extended time required to reach the peak current (t_m). (In the Manuscript, Page 12)

Supplementary Figure 17. Current-time transients obtained at predetermined potentials. j_m : peak current, t_m : time needed to achieve the peak current.

Supplementary Figure 15. (a) Typical voltage profiles of galvanostatic Zn deposition. (b) Voltage profiles of symmetrical cells at $1 \text{ mA cm}^{-2}/1 \text{ mAh cm}^{-2}$. (c) The calculated overpotential value. (d) Schematic diagram of Zn heterogeneous nucleation on the substrate based on the classical nucleation theory.

Typically, in the case of Zn plating at a constant current density, the voltage experiences a swift decline to a value below 0 V (vs. Zn^{2+}/Zn) and then levels off with a higher value. The nucleation overpotential (η_n) is defined as the onset voltage dip, which is related to the thermodynamic cost of forming a critical atomic cluster, while the mass-transfer controlled plateau overpotential (η_p) describes the growth process⁷. (In the Supporting Information, Page R13)

Supplementary Figure 13. SEM images of Zn deposits on Zn-Ti alloy after plating for (a) 15 s and (c) 30 s, along with the corresponding (b, d) histograms displaying the sizes of nuclei.

The evolution of nuclei size and number during the nucleation process was quantitatively investigated using StarDist, a publicly available plugin for Fiji (Fiji is just ImageJ). The histogram of the statistical data reveals an increase in the average size of Zn nuclei over time, accompanied by a slightly more scattered distribution, due to the generation of new Zn nuclei. (In the Supporting Information, Page R11)

The significant differences in the nucleation overpotentials are thought to be associated with the distinct affinities toward Zn⁴⁷. To gain a deeper insight into zincophilic behavior, the binding energies of Zn atom attached to the substrates were evaluated using density functional theory (DFT) calculations, with the selection of crystal planes guided by XRD results. As summarized in **Supplementary Fig. 16**, the calculated binding energy values between Zn atom and TiZn₁₆ (241) facet, (134) facet, and (313) facet are -1.07, -1.71, and -1.20 eV, respectively, which are more negative than those between Zn atom and Zn facets (-0.26, -0.66 and -0.85 eV). This indicates that TiZn₁₆ IMCs could potentially serve as nucleation sites, which endow the Zn-Ti alloy with high

zincophilicity to facilitate Zn deposition, thus leading to a reduced nucleation overpotential and a lower energy barrier. (In the Manuscript, Page 11)

Supplementary Figure 16. (a) Optimal models of Zn atom adsorption on Zn (002) facet, Zn (100) facet, Zn (101) facet, TiZn₁₆ (241) facet, TiZn₁₆ (134) facet and TiZn₁₆ (313) facet. (b) Calculated binding energies of Zn atom with different crystal facets.

The selection of crystal facets is based on the three most prominent diffraction peaks identified in the XRD results. In the schematic diagram of the optimized model, Zn and Ti atoms are colored in gray and blue, respectively, while the adsorbed Zn atom is highlighted in orange. (In the Supporting Information, Page R16)

3. It is challenging for me to correlate the model results with the experiment results. I assume that the numerical model should provide insights for understanding the Zn deposition. However, I only see some superficial descriptions regarding the morphology on some time points and also confirm with the SEM figure. If that is the case, what is the point of employing the model? I expect a more thorough discussion from the numerical model to help improve the understanding (of course, we need to make sure that the model is correct).

Response:

Thanks for the reviewer's valuable suggestion. We sincerely apologize for any inconvenience caused by our inappropriate descriptions and insufficiently in-depth discussions that may have affected your reading experience. Therefore, we have rewritten the "Favorable Nucleation and Growth Models" section, and have also incorporated additional details that are critical to the integrity of this part. In this section, we firstly examined the morphological evolution during the nucleation

and growth processes, as illustrated by the time-series SEM images (**Fig. 3a-b** and **Supplementary Fig. 12-13**). For bare Zn, Zn nuclei randomly appear at the limited nucleation sites, leaving behind widely uncovered regions. Subsequently, Zn^{2+} exhibits a preference to deposit onto existing nuclei, leading to the pronounced protrusions. While for Zn-Ti alloy, as the deposition moves forward, the number and size of Zn nuclei increase simultaneously. The abundant Zn nuclei then navigate the ensuing growth process. As neighboring particles come into contact with each other, the deposited Zn particles gradually merge together and cover the whole surface of the substrate. To elucidate the observed disparities in deposition morphology, we analyzed the voltage profiles obtained from galvanostatic electrodeposition (**Supplementary Fig. 15**). The Zn-Ti alloy demonstrates a small nucleation overpotential, suggesting a lower energy barrier. The distinctions in nucleation overpotentials can be attributed to zincophilicity, whereby $TiZn_{16}$, possessing a more negative binding energy, could potentially serve as nucleation sites to facilitate the process (**Supplementary Fig. 16**). More importantly, the much-reduced overpotential difference (gap between nucleation and plateau overpotentials) signifies that, for the Zn-Ti alloy, nucleation is more prone to occur compared to bare Zn (**Fig. 3d**). Subsequently, we employed chronoamperometry to investigate the initial nucleation model (**Fig. 3e** and **Supplementary Fig. 17**). The normalized dimensionless transient for bare Zn is well-fitted by the 3DI model, while Zn-Ti alloy follows a hybrid 3DI+3DP model. The 3DI mode implies that nuclei appear almost instantaneously at all possible but limited sites and then grow, whereas the partially progressive mode signifies nucleation sites being incrementally activated, endowing the Zn-Ti alloy with abundant Zn nuclei. These model results align well with the observed morphological disparities during the initial deposition process (**Fig. 3a**). Evidently, the nucleation modes will inevitably influence the local electrochemical environment, and thus the subsequent Zn deposition. The growth process following nucleation was simulated using COMSOL Multiphysics (**Fig. 3f-g**). The Zn nuclei formed at the early stage will interfere with the electric field and ion flux distribution at the interface (**Supplementary Fig. 18**). Since the priority of Zn growth depends on this distribution, the abundant and evenly distributed Zn nuclei on Zn-Ti alloy can diminish the heterogeneity of further growth, thereby developing into a relatively flat deposited layer (**Supplementary Fig. 19**). These simulated results are in accordance with the morphological

evolution depicted in **Fig. 3b**. Based on the above observations and analyses, the nucleation and growth mechanisms were deduced schematically in **Fig. 3h**, where solid circles with centers represent "nucleation" and dashed circles symbolize "growth".

Changes:

Obviously, the nature of metallic substrate has a profound influence on Zn deposition, prompting the question of how Zn-Ti alloy interacts with and regulates the deposition behavior. The morphological evolution of Zn deposits over time was firstly examined by ex-situ SEM imaging. The initial state of Zn deposition is depicted in **Fig. 3a**, and it is noteworthy that the deposits obtained in a short time are primarily determined by the nucleation process. It can be observed that Zn^{2+} tends to nucleate randomly on bare Zn. With the proceeding of deposition, these sparse nuclei continue to grow and coalesce, resulting in pronounced protrusions and leaving behind widely uncovered regions. Subsequent Zn^{2+} deposition on existing Zn nuclei induces inhomogeneity in the growth process, which can be traced back to the uneven distribution of nucleation sites, consequently intensifying the inclination towards localized Zn deposition (**Fig. 3b**). Moreover, the morphology of Zn deposits transforms from blocky to mossy, potentially aggravating the corrosion reactions (**Supplementary Fig. 12**)⁴³. In contrast, the deposits on Zn-Ti alloy exhibit a particulate-like morphology, as revealed by time-series SEM images. Similar-sized Zn particles nucleate uniformly at the initial stage of deposition. As the deposition progresses, the size of Zn nuclei increases, along with a rise in their quantity, suggesting a time-dependent progressive feature (**Supplementary Fig. 13**). The abundant Zn nuclei guide the subsequent growth process, and when adjacent nuclei come into contact with each other, they gradually merge into a dense Zn deposition layer that covers the whole surface of the substrate. The height characteristics of Zn deposits were then examined using atomic force microscopy (AFM). Notably, even after deposition for 60 min, the upper surface of Zn-Ti alloy consistently maintains a low roughness, while the deposits on bare Zn displays an undulating topography with distinct slopes whose height difference exceeds 3.8 μm (**Fig. 3c**). Evidently, the Zn deposition on Zn-Ti alloy reveals a homogeneous particulate-like deposition mode, quite different from the uneven dendrite-like deposition mode observed on bare Zn. Of particular note is the evident differences between the two morphologies right from the early stage of deposition, implying the

existence of a critical factor in the nucleation process that distinguishes the deposition modes. In addition, the stripping process following plating was also investigated, which holds equal importance⁴⁴. In contrast to the uniformly stripped surface observed on Zn-Ti alloy, loosely distributed "dead Zn" and deep pits are distributed on the surface of bare Zn (**Supplementary Fig. 14**). In fact, the inhomogeneous plating/stripping behavior will intertwine and recur over extended cycles, ultimately leading to interfacial instability and a decline in battery performance. These findings emphasize the significance of initial deposition quality for the subsequent plating/stripping reversibility, as discussed below. (In the Manuscript, Page 9)

According to classical nucleation theory, the formation of a new phase needs to overcome an energy barrier, which can be reflected electrochemically⁴⁵. Specifically, a detailed examination of the characteristic overpotential parameters will enhance the understanding of this thermodynamically related process, as depicted in **Supplementary Fig. 15**. The voltage profile obtained during galvanostatic deposition reveals an apparent voltage dip at the onset of electrodeposition on bare Zn, measuring 51 mV. In contrast, the Zn-Ti alloy displays a lower nucleation overpotential of only 24 mV, indicative of a reduced energy barrier. Note that there are minimal disparities in the plateau overpotentials, suggesting that the substrate primarily influences nucleation rather than growth process⁴⁶. The significant differences in the nucleation overpotentials are thought to be associated with the distinct affinities toward Zn⁴⁷. To gain a deeper insight into zincophilic behavior, the binding energies of Zn atom attached to the substrates were evaluated using density functional theory (DFT) calculations, with the selection of crystal planes guided by XRD results. As summarized in **Supplementary Fig. 16**, the calculated binding energy values between Zn atom and TiZn₁₆ (241) facet, (134) facet, and (313) facet are -1.07, -1.71, and -1.20 eV, respectively, which are more negative than those between Zn atom and Zn facets (-0.26, -0.66 and -0.85 eV). This indicates that TiZn₁₆ IMCs could potentially serve as nucleation sites, which endow the Zn-Ti alloy with high zincophilicity to facilitate Zn deposition, thus leading to a reduced nucleation overpotential and a lower energy barrier. Another noteworthy observation is that the voltage profile of Zn-Ti alloy exhibits a flat voltage turning without any noticeable voltage spike, which contrasts sharply with the response observed on bare Zn. Typically, the difference between two characteristic overpotentials is

regarded as an indicator for assessing the gap of driving forces between nucleation and growth processes⁴⁸. As semi-quantitatively shown in **Fig. 3d**, the large energy difference between the Gibbs free energy for Zn nucleation ($\Delta G_{\text{nucleation}}$) and growth (ΔG_{growth}) on bare Zn partly restricts nucleation while favoring the growth process. As a result, Zn^{2+} tends to deposit onto existing Zn nuclei, which may develop into dendrites as capacity increases⁴⁹. In contrast, the TiZn_{16} IMCs with high zincophilicity reduce the $\Delta G_{\text{nucleation}}$ to the same level of ΔG_{growth} , thereby promoting more and denser Zn nuclei and resulting in homogeneous Zn deposition. (In the Manuscript, Page 11)

Since the electrocrystallization of new phases is a strong function of overpotentials and can be directly reflected by current transients, chronoamperometry (CA) was employed to further probe the kinetics of Zn nucleation and growth. All the transients exhibit a peak current (j_m), indicating the typical three-dimensional (3D) nucleation process with hemispherical diffusion control (**Supplementary Fig. 17**)⁵⁰. Initially, the rise in current reflects the expansion of electroactive area, owing to the growth in size of existing nuclei and the generation of new nuclei on the surface⁵¹. Subsequently, the current decreases and approaches a steady state due to the overlapping of neighboring nuclei and their diffusion zones, indicating a diffusion-controlled growth process⁵². It is evident that the Zn-Ti alloy demonstrates a slower and more progressive nucleation process in comparison to bare Zn, as substantiated by the extended time required to reach the peak current (t_m). Furthermore, the current-time transients were normalized based on j_m and corresponding t_m , and compared with the Scharifker-Hills model⁵². Based on this theory, nucleation occurs in two modes, instantaneous (I) or progressive (P), depending on whether new nuclei emerge instantaneously at the beginning or progressively over time, and the mathematical expressions are described by:

$$3\text{DI: } \left(\frac{j}{j_m}\right)^2 = 1.9542 \left(\frac{t}{t_m}\right)^{-1} \left\{1 - \exp\left[-1.2564 \left(\frac{t}{t_m}\right)\right]\right\}^2 \quad (1)$$

$$3\text{DP: } \left(\frac{j}{j_m}\right)^2 = 1.2254 \left(\frac{t}{t_m}\right)^{-1} \left\{1 - \exp\left[-2.3367 \left(\frac{t}{t_m}\right)^2\right]\right\}^2 \quad (2)$$

The normalized dimensionless transients are shown in **Fig. 3e**, revealing that the nucleation on bare Zn follows closely the response predicted for 3DI, that is, the nucleation sites are relatively limited and become exhausted at early stage of the process. Conversely, the nucleation behavior observed on Zn-Ti alloy follows somewhat a hybrid model, that is, 3DI for $t < t_m$ as well as 3DP and

diffusion-controlled growth for $t > t_m$. In this scenario, nucleation sites are progressively activated and the process is accompanied with the nuclei growth, resulting in smaller particle sizes typically. These model results coincide well with the morphological differences observed during initial Zn deposition. The distinctions in nucleation modes will inevitably exert the localized electrochemical environment, thus shaping the subsequent growth behavior. (In the Manuscript, Page 12)

Additionally, finite element simulations using COMSOL Multiphysics were conducted to further understand the post-nucleation growth process. Hemispheres were employed to mimic Zn nuclei structure based on the preceding SEM observations. The initial modeling, depicted in **Fig. 3f**, illustrates that Zn nuclei of varying sizes are evenly distributed on Zn-Ti alloy, while block-like deposits are randomly scattered on bare Zn. Initially, the electrolyte current exhibits a discernible inclination to flow towards the Zn nuclei, with this tendency being more pronounced for bare Zn due to limited nucleation sites (**Supplementary Fig. 18**). Upon extending the simulation time, Zn^{2+} accumulates onto existing Zn nuclei along the trajectory defined by current vector, intensifying the propensity for localized deposition and consequently resulting in prominent protrusions (**Fig. 3g** and **Supplementary Fig. 19**). In contrast, the abundant Zn nuclei present on Zn-Ti alloy interact with the electric field and ion flux, diminishing the inhomogeneity of Zn growth. As deposition progresses, the gaps between Zn nuclei are gradually filled and the surface becomes smoother, in accordance with the experimental findings. (In the Manuscript, Page 13)

Building on the aforementioned observations and analysis, the potential nucleation and growth mechanisms were schematically summarized in **Fig. 3h**. For bare Zn following the 3DI model, Zn nuclei emerge randomly on the limited nucleation sites within a short time, and they tend to grow in their former positions without the formation of new nuclei due to the elevated nucleation barrier. The subsequent growth process is driven by the 3D volumetric extension of such structure, intensifying the heterogeneity of deposition, and thus the final appearance exhibits pronounced protrusions. For Zn-Ti alloy following the 3DI+3DP model, $TiZn_{16}$ IMCs with high zincophilicity could provide more nucleation sites that undergo progressive activation. The evenly distributed Zn nuclei guide the further growth, thereby yielding the relatively flat Zn deposition layer. In brief, Zn-Ti alloy was found to be effective in promoting favorable nucleation and growth processes, which are crucial for

highly reversible AZBs. (In the Manuscript, Page 14)

Fig. 3 | Mechanisms of Zn nucleation and growth. Time-series SEM images of Zn **a** nucleation and **b** growth. **c** AFM height images after Zn deposition for 60 min. **d** Semi-quantitative description of Gibbs free energy during Zn deposition. **e** Experimental dimensionless transients in comparison with the theoretical 3D nucleation models. Simulated **f** initial nuclei models and **g** final Zn deposition models after 240 s of plating. **h** Schematic illustration of the Zn nucleation and growth mechanisms.

4. In Fig. 3 (d) and (e), how many cycles have been there? Have the authors considered the stripping process as well?

Response:

We appreciate the valuable feedback from the reviewer. Indeed, **Fig. 3f-g** (with revised image order) presents the simulated results obtained through COMSOL Multiphysics 5.6 software, with the aim of investigating the post-nucleation growth process. **Fig. 3f** displays the simulated initial Zn nuclei models, with modeling referencing the preceding SEM observations (**Fig. 3a**). **Fig. 3g**

illustrates the final Zn deposition models after 240 s of plating. In addition, we have included numerical simulation results of the electrolyte current density vectors (indicated by arrows) to reflect the impact of initial Zn nuclei on the electric field and ion flux distribution (**Supplementary Fig. 18**). The intermediate models during the simulation process were also provided (**Supplementary Fig. 19**).

Since the Zn plating/stripping process alternately occurs during cell operation, the stripping process inevitably exerts an influence on the subsequent plating process as well as the overall cycling performance of the cell. Recognizing the significance of stripping process, we have included additional SEM images capturing the stripping process (**Supplementary Fig. 14**). Prior to this, we observed that the voltage profiles during the stripping process exhibit two distinct stages: a stable voltage stage and an increasing voltage stage. Previous research reports that the initial stable voltage corresponds to the stripping of pre-deposited Zn, which is relatively facile (*Nature Communications*, 2022, 13, 3699; *Joule*, 2019, 3, 485-502). Once the pre-deposited Zn that is readily strippable is fully stripped, the stripping process shifts to the substrate, resulting in a voltage spike, as supported by previous studies (*Energy Storage Materials*, 2023, 60, 102827). Clearly, bare Zn exhibits a short stable voltage range during the stripping process, followed by a noticeable voltage spike. This indicates the incomplete stripping of pre-deposited Zn and the consumption of substrate. The corresponding SEM images corroborate this inference, revealing a substantial presence of "dead Zn" and deep pits on the surface of bare Zn after stripping. In contrast, the voltage profile of Zn-Ti alloy remains stable, and its surface appears smoother after stripping, highlighting the benefits of Zn-Ti alloy not only for plating but also for the stripping process.

Response:

Additionally, finite element simulations using COMSOL Multiphysics were conducted to further understand the post-nucleation growth process. Hemispheres were employed to mimic Zn nuclei structure based on the preceding SEM observations. The initial modeling, depicted in **Fig. 3f**, illustrates that Zn nuclei of varying sizes are evenly distributed on Zn-Ti alloy, while block-like deposits are randomly scattered on bare Zn. Initially, the electrolyte current exhibits a discernible inclination to flow towards the Zn nuclei, with this tendency being more pronounced for bare Zn due

to limited nucleation sites (**Supplementary Fig. 18**). Upon extending the simulation time, Zn^{2+} accumulates onto existing Zn nuclei along the trajectory defined by current vector, intensifying the propensity for localized deposition and consequently resulting in prominent protrusions (**Fig. 3g** and **Supplementary Fig. 19**). In contrast, the abundant Zn nuclei present on Zn-Ti alloy interact with the electric field and ion flux, diminishing the inhomogeneity of Zn growth. As deposition progresses, the gaps between Zn nuclei are gradually filled and the surface becomes smoother, in accordance with the experimental findings. (In the Manuscript, Page 13)

Supplementary Figure 18. Numerical simulation results of the electrolyte current density vectors on (a) bare Zn and (b) Zn-Ti alloy.

The direction of the electrolyte current density vector coincides with the direction of the electric field, which also corresponds to the direction of Zn^{2+} flux. (In the Supporting Information, Page R18)

Supplementary Figure 19. The morphological evolution of Zn deposition on (a) bare Zn and (b) Zn-Ti alloy obtained by COMSOL simulations.

In addition, the stripping process following plating was also investigated, which holds equal importance⁴⁴. In contrast to the uniformly stripped surface observed on Zn-Ti alloy, loosely distributed "dead Zn" and deep pits are distributed on the surface of bare Zn (**Supplementary Fig. 14**). In fact, the inhomogeneous plating/stripping behavior will intertwine and recur over extended cycles, ultimately leading to interfacial instability and a decline in battery performance. (In the Manuscript, Page 10)

Supplementary Figure 14. SEM images of (a-c) bare Zn and (d-f) Zn-Ti after 30 min of plating, 60 min of plating and 60 min of plating followed by 60 min of stripping. (g) Voltage profiles of symmetrical cells at $1 \text{ mA cm}^{-2}/1 \text{ mAh cm}^{-2}$. (h) Schematic diagram of Zn plating and stripping on bare Zn and Zn-Ti alloy.

The voltage profiles observed during the subsequent stripping process can be divided into two parts: the stable voltage stage and the increasing voltage stage. The initial stable voltage corresponds to the stripping of pre-deposited Zn from the former process, which is easy to proceed⁵. Once the strippable pre-deposited Zn is depleted, the stripping process shifts to the substrate, resulting in a voltage spike⁶. Clearly, bare Zn exhibits a short stable voltage stage during this process, followed by a noticeable voltage spike. This indicates the incomplete stripping of pre-deposited Zn (also known as "dead Zn" formation) and the consumption of substrate, as further confirmed by the SEM images. In contrast, the surface of Zn-Ti alloy appears relatively smooth after stripping, highlighting the benefits of Zn-Ti alloy not only for plating but also for the stripping process. (In the Supporting Information, Page R12)

Review #3

This finding work demonstrated impressive results. The grain boundary engineering of Zn with Ti alloy is a simplified method but yield excellent results for Zn suppression. The well-designed experiment with the insight characterization reveals the novel finding in this field. Please check on my comments:

Response:

Thank you very much for your thorough review and constructive suggestions regarding our manuscript. We are honored that the enhancement in electrochemical stability brought by grain boundary engineering and the further exploration of Zn nucleation and growth mechanism can obtain such a positive evaluation. We have carefully considered the issues mentioned, and have made necessary supplements and corrections as per the reviewer's suggestions.

1. Why the author chose pure zinc and titanium in a weight ratio of 99.5:0.5 for this work. Is it an optimized condition? Is there any effect from grain size?

Response:

Thanks for the reviewer's reminding. We apologize for the simplistic description of element proportions selection. In order to ensure the efficient utilization of the Zn metal anode (i.e., to maximize the content of electrochemically active Zn), the addition of Ti should be kept as minimal as possible. Simultaneously, maintaining an adequate quantity of Ti is necessary to guarantee the effective presence of Ti-containing intermetallic compounds at the grain boundaries. Additionally, it is worth noting that as the Ti content increases, the form of TiZn_{16} transitions from a fine lamellar structure with continuous distribution at grain boundaries to a coarse dendritic structure with random distribution (*Microstructural Evolution and Mechanical Properties of Zn-Ti Alloys for Biodegradable Stent Applications*, M.S. Thesis, Michigan Technological University, 2017). By integrating the above analysis and referring to the phase diagram, we have opted for an empirical Zn-Ti weight ratio of 99.5:0.5. Under this condition, according to the Zn-Ti binary phase diagram (*Acta Materialia*, 2006, 54, 4977-4997), the cooling process undergoes peritectic and eutectic reaction, eventually leading to the crystallization of α -Zn phase and Ti-containing intermetallic phase. The enrichment of Ti-containing intermetallic compounds at the grain boundaries restrains the growth of grain, leading

to a decrease in the grain size of α -Zn phase (*Corrosion Science*, 2014, 89, 286-294).

Changes:

Based on the phase diagram, the utilization ratio of Zn anode can be maximized within the most Zn-rich region. Within this range, to ensure the presence of sufficient Ti-containing IMCs and to avoid dendritic structures, an empirical Zn-Ti weight ratio of 99.5:0.5 was employed. In this scenario, the alloy undergoes sequential phase transitions during the cooling process¹, ultimately crystallizing into a Zn-rich solid solution and TiZn₁₆ IMCs. (In the Supporting Information, Page R2)

2. How the intensity peak of Zn₄SO₄(OH)₆·H₂O in XRD pattern reflects the HER reaction, please explain and What state of sample is used to measure XRD?

Response:

Thanks for the reviewer's comment. According to the Pourbaix diagram of the Zn-H₂O system, the Zn metal is thermodynamically unstable in weakly acidic aqueous electrolytes, resulting in the HER-induced corrosion reactions (*Energy & Environmental Science*, 2021, 14, 5669-5689). During this process, hydrogen ions in the electrolytes are consumed, causing a localized increase in pH at the Zn/electrolyte interface. This, in turn, leads to the formation of insoluble by-products, which can be described by:

Therefore, the diffraction peak intensity of Zn₄SO₄(OH)₆·xH₂O can typically serve as an indicator to reflect the HER.

We apologize for not specifying the preparation process of the sample intended for testing. Considering that the HER is a slow process evolving over time, we conducted XRD testing on the Zn anode after subjecting it to 50 cycles, equivalent to 100 h of cycling, in a symmetrical cell (**Fig. 2c** and **Supplementary Fig. 9**). Since HER primarily occurs during the cathodic process, the electrode on deposition side was selected for a more representative characterization (*Nature Communications*, 2023, 14, 1828). Once the cycling test was completed, the electrode was promptly removed from the coin cell and thoroughly washed with deionized water to eliminate remaining electrolyte. The sample was then dried in a vacuum oven and utilized for various tests.

Changes:

Once the HER occurs, a significant amount of hydroxide ions is generated. The localized alkaline environment affects the chemical surroundings at the interface, thereby prompting the precipitation of zinc hydroxysulfate ($Zn_4SO_4(OH)_6 \cdot xH_2O$) from the electrolyte onto the surface of Zn anodes. The formation process can be represented as:

(In the Supporting Information, Page R6)

Considering that the HER primarily occurs during the cathodic process⁴, a typical deposition morphology was selected to identify the by-products. Following the completion of cycling in the symmetrical cell, the electrode on deposition side was immediately extracted and rinsed with deionized water to remove any residual electrolyte. The sample was then dried under vacuum and employed for various measurements, including SEM, XRD and XPS analysis. (In the Supporting Information, Page R7)

3. In Fig. 2e, how to obtain the ratio of phase recognize? Does it represent only for interested area? Can we use operando XRD technique to confirm the bulk properties?

Response:

We thank the reviewer for carefully reviewing our manuscript and for raising these valuable questions. The ratio of phase recognition was generated by EBSD analysis software, Channel 5, based on image identification technology, which is commonly used in the research of metallic materials. This software allows for obtaining microstructure information, such as phase distribution, grain orientation, and grain boundary characteristics, from any region of interest within the EBSD images. The ratio of phase recognition shown in **Fig. 2g** was derived from the image identification results in **Fig. 2f**, and the raw statistical data was presented in **Fig. R1** below.

Figure R1. Raw statistical data of phase recognition for (a-b) bare Zn and (c-d) Zn-Ti alloy before and after immersion.

To the best of our knowledge, there are currently no reports on the utilization of operando XRD to monitor the corrosion process, specifically the evolution of peak intensity associated with $\text{Zn}_4\text{SO}_4(\text{OH})_6 \cdot x\text{H}_2\text{O}$. On one hand, although the standard electrode potential of Zn^{2+}/Zn is lower than that of HER, the HER of metallic Zn in weakly acidic electrolytes does not occur vigorously due to its moderate hydrogen overpotential (*ACS Energy Letters*, 2021, 6, 1773-1785). Considering that the HER-induced corrosion reaction is time-dependent, continuous operation of an X-ray diffractometer for several days or even weeks to detect this slow process poses significant challenges and could potentially cause irreparable damage to the equipment. On the other hand, the randomness in the formation location of by-products also adds to the difficulty of detecting relevant diffraction peaks in a short time frame (*Joule*, 2023, 7, 1145-1175). Given the aforementioned considerations, we express our regret for not being able to furnish operando data in this regard. Instead, we have included ex-situ XRD patterns, SEM images, and optical photographs of Zn anodes immersed for varying durations to further validate the advantages of Zn-Ti alloy in corrosion inhibition. It should be noted that, unlike the results shown in **Fig. 2f** and **Supplementary Fig. 11**, where the surface by-products were removed to investigate the origin of corrosion, the present experiments aim to preserve the original

corrosion morphology to the greatest extent possible. As depicted in **Fig.2b** and **Supplementary Fig. 8**, the Zn-Ti alloy consistently maintains a compact surface with markedly reduced by-products, even when subjected to prolonged immersion for 10 days. These findings, serving as supplementary evidence, once again confirm the significant contribution of grain boundary engineering in stabilizing Zn anode.

Changes:

To further clarify the significance of GBE in inhibiting intergranular corrosion, EBSD analysis was performed to monitor the reaction process, in which the phase recognition was quantified on the basis of image identification. (In the Manuscript, Page 8)

Fig. 2 | Suppression of Parasitic Reaction through Grain Boundary Engineering. **a** Tafel plots. **b** Ex-situ XRD patterns of Zn anodes after immersion in ZnSO₄ electrolyte for different days. **c** Cross-sectional SEM images of Zn anodes after 50 cycles. Schematic diagram of **d** general corrosion and **e** intergranular corrosion. **f** EBSD IPF mapping images and corresponding **g** ratio of phase

recognition of Zn anodes at pristine state and after 24 h of immersion. CLSM imaging of corrosion morphologies on **h** bare Zn and **i** Zn-Ti alloy, and corresponding surface profiles extracted along the location marked. **j** “Reservoir” protocol for evaluating average Zn plating/stripping CE.

The propensity of the Zn anode to degrade under prolonged aging and cycling in mildly acidic electrolyte, owing to reactions with water solvent, results in the irreversible consumption of metallic Zn into by-products³⁹. To evaluate the shelf life, Zn anodes were immersed in ZnSO₄ electrolytes for varying durations. It is evident that for bare Zn, the diffraction peaks corresponding to Zn₄SO₄(OH)₆·3H₂O (PDF#39-0689) begin to appear after 1 day of immersion and become significant after 10 days, indicating severe corrosion reactions (**Fig. 2b**). The corresponding SEM and EDS mapping further corroborate this result, revealing discernible flake-like by-products and noticeable signs of corrosion (**Supplementary Fig. 8**). Conversely, the surface of Zn-Ti alloy displays minimal by-products, with the diffraction peaks of zinc hydroxysulfate remaining inconspicuous even after 10 days of immersion, demonstrating the excellent corrosion inhibition capability. (In the Manuscript, Page 7)

Supplementary Figure 8. (a) Optical photograph of the Zn anodes during immersion. SEM and corresponding EDS elemental mapping of (b-c) Zn-Ti alloy and (d-e) bare Zn after 10 days of immersion.

It is noteworthy that during the immersion process, the surface of Zn-Ti alloy exhibits only a few sporadic bubbles, in sharp contrast to the evident bubble accumulation observed on bare Zn. (In the Supporting Information, Page R6)

4. Could the author provide the corrosion rate from Tafel plot?

Response:

Thanks for the reviewer's comment. In our previous manuscript, we utilized Tafel extrapolation method in the strong polarization region of polarization curves to estimate the corrosion current densities of Zn anodes (**Fig. 2a**), a widely applied approach for qualitative analysis of corrosion phenomena (*Joule*, 2023, 7, 1145-1175). The corresponding corrosion rate can be described as follows:

$$v = \frac{36000M}{nF} i_{\text{corr}} \quad (2)$$

where i_{corr} represents the corrosion current density, M the formula weight, n the electron number, F the Faraday's constant. Consequently, the corrosion rate of bare Zn is determined to be 42.65 g m⁻² h⁻¹, while that of Zn-Ti alloy is 23.36 g m⁻² h⁻¹. In the case of Zn corrosion, the corrosion rate is directly proportional to the corrosion current density. In order to facilitate comparison across different literature sources, we ultimately opt for the commonly used corrosion current density rather than corrosion rate as the metric for this study.

5. Why the CV curve in Supplementary Figure 9. was employed by using Ag/AgCl instead of Zinc reference.

Response:

Thank you for your valuable feedback. In various electrochemical experiments, such as LSV, Tafel, and CV, researchers commonly employ a three-electrode system with a stable reference electrode (*Nature Communications*, 2023, 14, 76; *Advanced Materials*, 2023, 35, 2211961). This choice is made because in a two-electrode system, the counter electrode also functions as the reference electrode, and when current passes through it, polarization occurs, resulting in potential drift and inaccurate measurements (*Journal of The Electrochemical Society*, 2022, 169, 070509).

However, a three-electrode system allows the working electrode to establish a current circuit with the counter electrode and a voltage circuit with the reference electrode. Since the applied voltage represents the potential difference between working electrode and reference electrode, it is essential for the reference electrode to exhibit stable and reproducible potential characteristics. According to the Nernst equation, the potential of the Ag/AgCl electrode (with saturated KCl as the electrolyte) is predominantly governed by chloride ions:

$$\varphi = \varphi' - 0.0592 \log c_{\text{Cl}^-} \quad (3)$$

thus ensuring relative stability. Moreover, its high exchange current density contributes to favorable reversibility. In contrast, the Zn electrode itself displays thermodynamic activity in weak acidic electrolytes, resulting in an unstable electrode potential, which renders it unsuitable for serving as a reference electrode. In summary, for the sake of experimental accuracy, we have opted for the more stable Ag/AgCl electrode instead of Zn electrode as the reference.

6. It might be perfect to support the change of structure during cycling test and verify the by-product from parasitic reactions at Zn surface by ex-situ XPS along with XRD and SEM.

Response:

We thank the reviewer for helpful suggestion, and have included the XPS results of Zn anodes after cycling. As presented in **Supplementary Fig. 9**, the S 2p XPS spectra clearly show that the peaks corresponding to SO_4^{2-} were detected on the surface of Zn anodes after cycling, with a significantly higher intensity observed on bare Zn compared to Zn-Ti alloy. Combined with XRD and SEM analyses, the by-product was identified as zinc hydroxysulfate. In addition, the intensity of Zn 2p XPS peak for bare Zn was found to be lower than that of Zn-Ti alloy, which can be attributed to the substantial accumulation of flake-like by-products on the surface, thereby affecting the signal of underlying Zn.

Changes:

Supplementary Figure 9. Top-view SEM images of (a) Zn-Ti alloy and (b) bare Zn after 50 cycles at $1 \text{ mA cm}^{-2}/1 \text{ mAh cm}^{-2}$. (c) The EDS elemental mapping of deposits on bare Zn. (d) Zn 2p, (e) S 2p XPS spectra and (f) XRD patterns of Zn anodes after cycling.

The top-view SEM observations indicate the existence of hexagonal flake-like species on the surface of bare Zn after cycling, and the corresponding EDS element mapping confirms that the predominant constituents of the by-products are Zn, S, and O. Further XPS analysis reveals that the peaks corresponding to SO_4^{2-} at approximately 168.9 eV are detected on the surface of Zn anodes after cycling, with a significantly higher intensity observed on bare Zn compared to Zn-Ti alloy. Combining the XRD results, the by-product is identified as $\text{Zn}_4\text{SO}_4(\text{OH})_6 \cdot \text{H}_2\text{O}$ (PDF#39-0690). (In the Supporting Information, Page R7)

7. Favorable Nucleation and Growth Models used in this work can predict only at early state of Zn deposition or continue deposited Zn on substrate. Since Zn anode has deposition and dissolution during plating/stripping process, how about the explanation of the dissolution of Zn on Zn-Ti during stripping. Is there any benefit of Zn-Ti after Zn dissolution for the next cycle of plating process.

Response:

We sincerely appreciate the valuable feedback provided by the reviewer. Since the Zn plating/stripping process alternately occurs during battery operation, the stripping process inevitably influences the subsequent plating process. It is widely accepted that a smooth surface facilitates the homogeneous distribution of electric field and ion flux at the interface. Thus, the dense deposition layer resulting from the favorable nucleation and growth model is anticipated to promote a well-organized stripping process. To substantiate this hypothesis, we have included additional SEM images capturing the stripping process following plating. Prior to this, we observed that the voltage profiles during the stripping process exhibit two distinct stages: a stable voltage stage and an increasing voltage stage. Previous research reports that the initial stable voltage corresponds to the stripping of pre-deposited Zn, which is relatively facile (*Nature Communications*, 2022, 13, 3699; *Joule*, 2019, 3, 485-502). Once the pre-deposited Zn that is readily strippable is fully stripped, the stripping process shifts to the substrate, resulting in a voltage spike, as supported by previous studies (*Energy Storage Materials*, 2023, 60, 102827). As depicted in **Supplementary Fig. 14**, bare Zn exhibits a short stable voltage range during the stripping process, followed by a noticeable voltage spike. This indicates the incomplete stripping of pre-deposited Zn and the consumption of substrate (i.e., the remaining Zn foil). The corresponding SEM images corroborate this inference, revealing a substantial presence of "dead Zn" and deep pits on the surface of bare Zn after stripping. In contrast, the voltage profile of Zn-Ti alloy remains stable, and its surface appears smoother after stripping, highlighting the benefits of the Zn-Ti alloy not only for plating but also for the stripping process. In the following cycles, the plating/stripping behavior of Zn is generally similar to that of the first cycle. The uneven plating/stripping behavior of bare Zn will intertwine and repeat continuously throughout the cycling process, easily resulting in the deterioration of battery performance. Attractively, different from bare Zn, the Zn-Ti alloy consistently exhibits a flat and orderly uniform morphology, which signifies a significant improvement in the homogeneity of plating/stripping.

Changes:

In addition, the stripping process following plating was also investigated, which holds equal importance⁴⁴. In contrast to the uniformly stripped surface observed on Zn-Ti alloy, loosely

distributed "dead Zn" and deep pits are distributed on the surface of bare Zn (Supplementary Fig. 14). In fact, the inhomogeneous plating/stripping behavior will intertwine and recur over extended cycles, ultimately leading to interfacial instability and a decline in battery performance. These findings emphasize the significance of initial deposition quality for the subsequent plating/stripping reversibility, as discussed below. (In the Manuscript, Page 10)

Supplementary Figure 14. SEM images of (a-c) bare Zn and (d-f) Zn-Ti after 30 min of plating, 60 min of plating and 60 min of plating followed by 60 min of stripping. (g) Voltage profiles of symmetrical cells at $1 \text{ mA cm}^{-2}/1 \text{ mAh cm}^{-2}$. (h) Schematic diagram of Zn plating and stripping on bare Zn and Zn-Ti alloy.

The voltage profiles observed during the subsequent stripping process can be divided into two parts: the stable voltage stage and the increasing voltage stage. The initial stable voltage corresponds to the stripping of pre-deposited Zn from the former process, which is easy to proceed⁵. Once the

strippable pre-deposited Zn is depleted, the stripping process shifts to the substrate, resulting in a voltage spike⁶. Clearly, bare Zn exhibits a short stable voltage stage during this process, followed by a noticeable voltage spike. This indicates the incomplete stripping of pre-deposited Zn (also known as "dead Zn" formation) and the consumption of substrate, as further confirmed by the SEM images. In contrast, the surface of Zn-Ti alloy appears relatively smooth after stripping, highlighting the benefits of Zn-Ti alloy not only for plating but also for the stripping process. (In the Supporting Information, Page R12)

8. The value of y axis on supplementary Figure 11 might not be correct.

Response:

We sincerely appreciate the reviewer for thorough examination and valuable feedback. After carefully examining the original data, we have confirmed that the value of y-axis is indeed accurate. Similar studies in the literature (*Energy & Environmental Science*, 2021, 14, 5563-5571; *Energy & Environmental Science*, 2021, 14, 4077-4084), employing the chronoamperometry technique to investigate the nucleation mechanism, have reported current densities within the same order of magnitude as those obtained in our study (**Fig. R2**).

Figure R2. (a) Current-time curves in the normal electrolyte and Zn-Ce electrolyte; (b) Current-time transients measured at a series of potentials in 0.5 M Zn²⁺ electrolytes.

9. Fig. 3a, it is not clear between nucleation and growth.

Response:

We greatly appreciate the reviewer's reminder and extend our apologies for any errors in the schematic illustration due to our oversight. To rectify this, we have reworked the schematic illustration (in two different views), incorporated appropriate annotations and provided further elucidations within the manuscript, for better clarity and ease of understanding. Indeed, building upon the preceding observations and analysis, the distinction primarily lies in the partially progressive mode endowing Zn-Ti alloy with abundant nucleation sites, which are gradually activated over time. Conversely, nucleation sites on bare Zn are depleted initially, with deposition predominantly manifesting through the growth of limited Zn nuclei. The nucleation and growth mechanisms were finally deduced in **Fig. 3h** (with revised image order), where solid circles with centers represent "nucleation" and dashed circles symbolize "growth".

Changes:

Building on the aforementioned observations and analysis, the potential nucleation and growth mechanisms were schematically summarized in **Fig. 3h**. For bare Zn following the 3DI model, Zn nuclei emerge randomly on the limited nucleation sites within a short time, and they tend to grow in their former positions without the formation of new nuclei due to the elevated nucleation barrier. The subsequent growth process is driven by the 3D volumetric extension of such structure, intensifying the heterogeneity of deposition, and thus the final appearance exhibits pronounced protrusions. For Zn-Ti alloy following the 3DI+3DP model, TiZn_{16} IMCs with high zincophilicity could provide more nucleation sites that undergo progressive activation. The evenly distributed Zn nuclei guide the further growth, thereby yielding the relatively flat Zn deposition layer. In brief, Zn-Ti alloy was found to be effective in promoting favorable nucleation and growth processes, which are crucial for highly reversible AZBs. (In the Manuscript, Page 14)

Fig. 3 | Mechanisms of Zn nucleation and growth. Time-series SEM images of Zn **a** nucleation and **b** growth. **c** AFM height images after Zn deposition for 60 min. **d** Semi-quantitative description of Gibbs free energy during Zn deposition. **e** Experimental dimensionless transients in comparison with the theoretical 3D nucleation models. Simulated **f** initial nuclei models and **g** final Zn deposition models after 240 s of plating. **h** Schematic illustration of the Zn nucleation and growth mechanisms.

10. Fig 3c, how to calculate Gib free energy? And also for $\Delta G_{nucleation}$ and ΔG_{growth} ? Please provide the information.

Response:

Thanks for the reviewer’s constructive comment. The semi-quantitative description of Gibbs free energy during Zn deposition presented in **Fig. 3d** (with revised image order) was drawn in reference to the research by Nazar et al. (*Joule*, 2022, 6, 1103-1120), where the magnitudes of Gibbs free energy were based on the characteristic overpotentials, namely nucleation overpotential and plateau overpotential (**Supplementary Fig. 15**). In fact, electrodeposition of new phase is an

electrochemical process where the energy required for nucleation and growth originates from the interfacial electric field. When a constant current is applied, cathodic current is directed towards the surface charging, and nucleation only occurs when polarization reaches a specific threshold, followed by growth. The electrical energy consumed during nucleation and growth is directly proportional to the applied overpotential (*Journal of The Electrochemical Society*, 2013, 160, A662-A668):

$$\Delta G \propto zF\eta \quad (4)$$

Consequently, higher overpotentials demand higher Gibbs free energy.

Changes:

Another noteworthy observation is that the voltage profile of Zn-Ti alloy exhibits a flat voltage turning without any noticeable voltage spike, which contrasts sharply with the response observed on bare Zn. Typically, the difference between two characteristic overpotentials is regarded as an indicator for assessing the gap of driving forces between nucleation and growth processes⁴⁸. As semi-quantitatively shown in **Fig. 3d**, the large energy difference between the Gibbs free energy for Zn nucleation ($\Delta G_{\text{nucleation}}$) and growth (ΔG_{growth}) on bare Zn partly restricts nucleation while favoring the growth process. As a result, Zn^{2+} tends to deposit onto existing Zn nuclei, which may develop into dendrites as capacity increases⁴⁹. In contrast, the TiZn_{16} IMCs with high zincophilicity reduce the $\Delta G_{\text{nucleation}}$ to the same level of ΔG_{growth} , thereby promoting more and denser Zn nuclei and resulting in homogeneous Zn deposition. (In the Manuscript, Page 11)

Supplementary Figure 15. (a) Typical voltage profiles of galvanostatic Zn deposition. (b) Voltage profiles of symmetrical cells at $1 \text{ mA cm}^{-2}/1 \text{ mAh cm}^{-2}$. (c) The calculated overpotential value. (d) Schematic diagram of Zn heterogeneous nucleation on the substrate based on the classical nucleation theory.

Typically, in the case of Zn plating at a constant current density, the voltage experiences a swift decline to a value below 0 V (vs. Zn^{2+}/Zn) and then levels off with a higher value. The nucleation overpotential (η_n) is defined as the onset voltage dip, which is related to the thermodynamic cost of forming a critical atomic cluster, while the mass-transfer controlled plateau overpotential (η_p) describes the growth process⁷. Note that the plateau overpotential is generally lower than the nucleation overpotential, as the addition of Zn atoms to existing nuclei is more favorable than forming a stable atomic cluster⁸. In addition, the overpotential difference ($\Delta\eta$) is defined as the gap between nucleation overpotential and plateau overpotential. (In the Supporting Information, Page R13)

On the other hand, the formation of nuclei during electrocrystallization requires the consumption of electrical energy, which is proportional to the applied overpotential (external cause). When a constant current is applied, the cathodic current is directed towards surface charging, and

nucleation only occurs once the polarization reaches a certain threshold. Consequently, the higher the required overpotential, the larger the energy barrier, which is energetically unfavorable. (In the Supporting Information, Page R15)

11. *To compare the Zn growth on Zn-Ti in Fig. 3g, the magnification of SEM for Zn-Ti should be similar to Zn.*

Response:

We really appreciate the reviewer's constructive comment. Indeed, we observed significant differences in the size of Zn deposits on bare Zn and Zn-Ti alloy during the early nucleation stage, which became more pronounced as the plating process progressed. This disparity can be attributed to the 3DI mode observed on bare Zn, where Zn nuclei appear almost instantaneously and then grow together, resulting in larger deposit sizes. In contrast, the Zn-Ti alloy exhibits a partially progressive mode (3DI+3DP mode), where the growth of Zn nuclei and the emergence of new Zn nuclei occur simultaneously during the plating process (**Fig. 3b**), resulting in smaller Zn deposits and a uniform deposition. Regrettably, it is challenging to effectively showcase the morphological characteristics of the two distinct Zn deposits at the same scale. To address this, we have included SEM images at the same magnification in the Supplementary Information (**Supplementary Fig. 14**), and different scale bars have been employed in the Manuscript to facilitate a more comprehensive comparison.

Changes:

Supplementary Figure 14. SEM images of (a-c) bare Zn and (d-f) Zn-Ti after 30 min of plating, 60 min of plating and 60 min of plating followed by 60 min of stripping. (g) Voltage profiles of symmetrical cells at $1 \text{ mA cm}^{-2}/1 \text{ mAh cm}^{-2}$. (h) Schematic diagram of Zn plating and stripping on bare Zn and Zn-Ti alloy.

The size of Zn deposits on bare Zn and Zn-Ti alloy during the plating process exhibits a significant difference. To facilitate a better comparison, distinct scale bars have been utilized in the manuscript. (In the Supporting Information, Page R12)

12. How such a Zn-Ti can reduce the significant voltage drop at the beginning of Zn deposition. Please provide more explanation.

Response:

Thanks for the reviewer's helpful comments. Zn-Ti alloy was found to markedly reduce the

voltage drop at the beginning of Zn deposition, namely a significant decrease in nucleation overpotential (**Supplementary Fig. 15**), which can be attributed to the strong zincophilic nature of TiZn₁₆ intermetallic compounds. Typically, the Zn deposition initiates through heterogeneous nucleation on the surface of Zn metal anode substrate (**Fig. R3**). The characteristics of substrate, such as composition, morphology, and surface chemistry, all have direct impacts on the distribution of concentration/electric fields around the substrate, thus influencing the nucleation process. According to classical nucleation theory, in the case of heterogeneous nucleation, the critical nucleation energy is derived as (*Chemical Reviews*, 2021, 121, 5986-6056):

$$\Delta G_{\text{het}}^* = \frac{(2 - 3 \cos \theta + \cos^3 \theta)}{4} \Delta G_{\text{hom}}^* \quad (5)$$

where θ is the contact angle between Zn nucleus and the substrate. Considering the quantitative relationship between interfacial tension and contact angle, expressed as:

$$\cos \theta = \frac{\gamma_{\text{SE}} - \gamma_{\text{SN}}}{\gamma_{\text{NE}}} \quad (6)$$

When the binding energy between Zn nucleus and the substrate becomes more negative, indicating a zincophilic nature, γ_{NE} is reduced and thus $\cos \theta$ is increased. Consequently, the nucleation energy barrier is decreased, leading to nucleation occurring at a lower nucleation overpotential. Therefore, binding energy could serve as a rational descriptor of the zincophilicity of anode substrate, which is also reflected electrochemically in the nucleation overpotential (*Science Advances*, 2019, 5, eaau7728). We then conducted additional calculations to ascertain the binding energies between Zn atom and Zn/TiZn₁₆ facets, aiming to validate this hypothesis. DFT calculations reveal that, compared to the Zn facets, the TiZn₁₆ facets exhibit more negative binding energies, endowing the Zn-Ti alloy with higher zincophilicity and thus a reduced nucleation overpotential (**Supplementary Fig. 16**).

Figure R3. Schematic diagram of Zn heterogeneous nucleation on the substrate.

Changes:

The nucleation of Zn nuclei involves a highly intricate process. From a thermodynamic perspective, the stability of a Zn nucleus deposited on the electrically charged substrate is determined by the combination of bulk free energy, comprising both chemical and electrical, and the interfacial free energy arising from the generation of new interfaces, which can be expressed by⁹:

$$\Delta G_{\text{sys}} = \left(\Delta G_V + \frac{zF\eta}{V_m} \right) S_V r^3 + (\gamma_{\text{NE}} S_{\text{NE}} + \gamma_{\text{SN}} S_{\text{SN}} - \gamma_{\text{SE}} S_{\text{SN}}) r^2 \quad (2)$$

$$S_V = \frac{\pi}{3} (2 - 3 \cos \theta + \cos^3 \theta) \quad (3)$$

$$S_{\text{NE}} = 2\pi(1 - \cos \theta) \quad (4)$$

$$S_{\text{SN}} = \pi(1 - \cos^2 \theta) \quad (5)$$

where ΔG_V stands for the volume free energy change (from liquid phase to solid phase), z the number of electrons transferred, F the Faraday's constant, η the overpotential, V_m the molar volume of Zn, S_V the volume factor of spherical cap, S_{NE} the curved surface area factor, S_{SN} the bottom surface area factor, γ_{NE} the nucleus/electrolyte interfacial free energy, γ_{SN} the substrate/nucleus interfacial free energy, γ_{SE} the substrate/electrolyte interfacial free energy, θ the contact angle, and r the radius of curvature of the nucleus. By incorporating the quantitative relationship between interfacial tension and contact angle, known as the Young's equation:

$$\gamma_{\text{SN}} - \gamma_{\text{SE}} = -\gamma_{\text{NE}} \cos \theta \quad (6)$$

Equation (2) can be expressed as follows:

$$\Delta G_{\text{sys}} = \left(\Delta G_V + \frac{zF\eta}{V_m} \right) S_V r^3 + \gamma_{\text{NE}} (S_{\text{NE}} - S_{\text{SN}} \cos \theta) r^2 \quad (7)$$

Physically, nucleation is thermodynamically favored when the volumetric and interfacial tension contributions to the energetics of system overcome the critical free energy. Given $d\Delta G_{\text{sys}}/dr = 0$, the critical radius required to form a thermodynamically stable atomic cluster can be derived as:

$$r_{\text{crit}} = \frac{-2\gamma_{\text{NE}} V_m}{zF\eta + \Delta G_V V_m} \quad (8)$$

This indicates that, for the same volume free energy change, the driving force for nucleation is dominated by the bulk electrostatic energy. Here, η_0 is defined as the characteristic overpotential:

$$\eta_0 = \frac{\Delta G_V V_m}{zF} \quad (9)$$

and therefore, the dimensionless overpotential can be written as:

$$\hat{\eta} = \frac{\eta}{\eta_0} \quad (10)$$

By substituting Equation (8) into Equation (7), the critical free energy to heterogeneously form a nucleus is given by:

$$\Delta G_{\text{crit}} = \frac{16\pi\gamma_{\text{NE}}^3 (2 - 3 \cos \theta + \cos^3 \theta)}{3\Delta G_V^2 4(1 + \hat{\eta})^2} \quad (11)$$

and expressed in dimensionless form:

$$\Delta \hat{G}_{\text{crit}} = \frac{\Delta G_{\text{crit}}}{\Delta G_0} = \frac{(2 - 3 \cos \theta + \cos^3 \theta)}{4(1 + \hat{\eta})^2} \quad (12)$$

where ΔG_0 is the characteristic Gibbs free energy of the system, which also corresponds to the critical Gibbs free energy for homogeneous nucleation in the electrolyte ($\theta = 180^\circ$) at $\eta = 0$. Calculations demonstrate that, for the system with a small contact angle (internal cause), namely zincophilic substrate, the critical free energy is reduced and nucleation occurs at the overpotential of smaller magnitude. (In the Supporting Information, Page R13)

The significant differences in the nucleation overpotentials are thought to be associated with the distinct affinities toward Zn⁴⁷. To gain a deeper insight into zincophilic behavior, the binding energies of Zn atom attached to the substrates were evaluated using density functional theory (DFT) calculations, with the selection of crystal planes guided by XRD results. As summarized in **Supplementary Fig. 16**, the calculated binding energy values between Zn atom and TiZn₁₆ (241) facet, (134) facet, and (313) facet are -1.07, -1.71, and -1.20 eV, respectively, which are more negative than those between Zn atom and Zn facets (-0.26, -0.66 and -0.85 eV). This indicates that TiZn₁₆ IMCs could potentially serve as nucleation sites, which endow the Zn-Ti alloy with high zincophilicity to facilitate Zn deposition, thus leading to a reduced nucleation overpotential and a lower energy barrier. (In the Manuscript, Page 11)

Supplementary Figure 16. (a) Optimal models of Zn atom adsorption on Zn (002) facet, Zn (100) facet, Zn (101) facet, TiZn₁₆ (241) facet, TiZn₁₆ (134) facet and TiZn₁₆ (313) facet. (b) Calculated binding energies of Zn atom with different crystal facets.

The selection of crystal facets is based on the three most prominent diffraction peaks identified in the XRD results. In the schematic diagram of the optimized model, Zn and Ti atoms are colored in gray and blue, respectively, while the adsorbed Zn atom is highlighted in orange. (In the Supporting Information, Page R16)

13. Why the use of Zn-Ti anode can yield the high capacity and better rate performance than that of Zn anode? Please give an explanation.

Response:

Thank you for the constructive feedback from the reviewer. At a low rate of 0.2 A g⁻¹, the capacities of full cells assembled with both types of Zn anodes are similar (**Supplementary Fig. 31**). Zn-Ti//NH₄V₄O₁₀, however, showcases a slightly higher capacity, which can be attributed to the reduced voltage gap between redox peaks (cathodic peaks: 0.598 V vs. 0.557 V, 0.908 V vs. 0.881 V; anodic peaks: 0.802 V vs. 0.827 V, 1.069 V vs. 1.086 V) (**Supplementary Fig. 29**). Hence, when subjected to galvanostatic charge-discharge tests within the same voltage range (0.4-1.4 V vs. Zn²⁺/Zn), the full cell equipped with Zn-Ti alloy exhibits higher capacities (**Fig. R4**). The reduced voltage gap observed in Zn-Ti//NH₄V₄O₁₀ primarily originates from the diminished voltage hysteresis on the anode side (*Nature Communications*, 2020, 11, 829), as reflected by the voltage profiles of half cells presented earlier in the manuscript. Given that bare Zn exhibits more pronounced voltage hysteresis under high current densities (**Fig. 4b**), the resultant full cell

consequently displays sluggish rate performance. The improved rate capability of Zn-Ti//NH₄V₄O₁₀ stems from the enhanced kinetics on the anode side, as manifested in the lower and stable voltage hysteresis in half cells, as well as in the decreased charge-transfer resistance (low-frequency region) and steeper slope (high-frequency region) observed in electrochemical impedance spectroscopy analysis (**Supplementary Fig. 30**). To further validate the kinetic advantage of the Zn-Ti alloy, supplementary calculations related to activation energy were conducted (**Supplementary Fig. 24**). Notably, the Zn-Ti alloy exhibits lower charge-transfer resistance at all temperatures, confirming its fast electron transfer characteristic. The calculated activation energy for Zn-Ti alloy is 39.84 kJ mol⁻¹, which is lower than that of bare Zn (46.05 kJ mol⁻¹), confirming the accelerated reaction kinetics.

Figure R4. Voltage profiles of full cells at 0.2, 1 and 5 A g⁻¹.

Changes:

To obtain a deeper comprehension of the accelerated charge transfer, impedance measurements were conducted at incremental temperatures. Notably, Zn-Ti alloy exhibits consistently lower interfacial impedance across all temperature ranges (**Supplementary Fig. 24**). Accordingly, the activation energy was calculated using the Arrhenius equation, which reveals a reduction in activation energy from 46.05 kJ mol⁻¹ to 39.84 kJ mol⁻¹ when employing Zn-Ti alloy instead of bare Zn, confirming the improved reaction kinetics. (In the Manuscript, Page 15)

Supplementary Figure 24. Nyquist plots of symmetric cells using (a) bare Zn and (b) Zn-Ti alloy at different temperatures. (c) The fitted curves and the corresponding activation energies calculated by Arrhenius equation.

The desolvation of Zn^{2+} at the interface is commonly regarded as the rate-limiting step for Zn deposition, which determines the reaction kinetics throughout the process. The activation energy for desolvation was deduced through linear fitting, employing the Arrhenius equation¹⁰:

$$\frac{1}{R_{ct}} = A \exp\left(\frac{-E_a}{RT}\right) \quad (13)$$

where R_{ct} represents the charge-transfer resistance, A the pre-exponential factor, E_a the activation energy, R the gas constant, and T the temperature in Kelvin. The R_{ct} values at each temperature were estimated by extrapolating the corresponding semicircles in the high-frequency regions. (In the Supporting Information, Page R24)

Supplementary Fig. 29 illustrates the typical CV curves of full cells, unveiling comparable energy storage behavior, with distinct vanadium-related redox peaks. The distinction lies in the fact that, compared to the bare Zn counterpart, Zn-Ti/ $NH_4V_4O_{10}$ exhibits a diminished voltage gap between the cathodic and anodic peaks, coupled with an amplified current response. The

improvement in reaction kinetics is also corroborated by impedance analysis, which demonstrates reduced charge-transfer resistance and steeper slope in the relevant frequency regions (**Supplementary Fig. 30**)⁵⁶. As a result, the rate evaluation of full cell with Zn-Ti alloy presents the slightly enhanced capacity retention. (**Fig. 5a** and **Supplementary Fig. 31**). (In the Manuscript, Page 17)

The anodic (oxidation) peaks (0.802 V, 1.069 V) of Zn-Ti//NH₄V₄O₁₀ are lower than those of bare Zn//NH₄V₄O₁₀ (0.827 V, 1.086 V), and the cathodic (reduction) peaks of Zn-Ti//NH₄V₄O₁₀ (0.598 V, 0.908 V) are all higher than those of bare Zn//NH₄V₄O₁₀ (0.557 V, 0.881 V). These findings suggest a narrower voltage gap for Zn-Ti//NH₄V₄O₁₀, consistent with the smaller voltage hysteresis observed in half cells. (In the Supporting Information, Page R27)

In the case of the full cell incorporating Zn-Ti alloy, the steeper slope in the low-frequency region reflects the enhanced Zn²⁺ diffusion kinetics and the lower R_{ct} in the high-frequency region indicates the faster charge transfer. (In the Supporting Information, Page R28)

REVIEWERS' COMMENTS

Reviewer #1 (Remarks to the Author):

Recommendation: Acceptance after minor revisions noted.

The manuscript titled "Tailoring grain boundary stability of Zn-Ti alloy for long-lasting aqueous Zinc batteries" has commendably incorporated the feedback from reviewers, resulting in a significantly improved overall content and detailed explanations. Notably, this article presents insightful findings on the zinc metal deposition mechanism on Zn-Ti alloy in relation to the nucleation potential of zinc metal and the formation of nucleation sites with Zn-Ti alloy at grain boundaries. These insights are substantiated through electrochemical analysis coupled with voltage profiles and further detailed analysis. The revised main figures and supplementary figures (S13 to S17), featuring computational modeling, electrochemical interpretation, and ex-situ microscope images of the surface morphology of zinc alloy electrodes and their interpretation, have further enriched the results. Thus, the authors should address several concerns below and revised the manuscript before publication in Nature Communications.

1. Provide surface characterization data with CLSM or AFM imaging for both bare Zn and Zn-Ti alloy before the cycling process. This additional information will enhance the understanding of morphological changes by metallurgical engineering for Zn anode system.
2. Please include detailed information on the XPS analysis in the supplementary materials. This will aid readers in gaining a more thorough understanding of the characterization techniques employed in the research.
3. In Supplementary Figure 24, where the EIS results exhibit different shapes between Zn and Zn-Ti alloy, we suggest including an equivalent circuits in the supplementary materials to elucidate these variations. Additionally, consider explaining if the electrical resistivity was influenced by metallurgical processes, and support this with I-V curves for clarity.

Reviewer #2 (Remarks to the Author):

I am impressed by the thorough revision of the manuscript and satisfactory answer to reviewers' comments. I am now convinced that this is a top-quality piece of work that should be accepted.

Reviewer #3 (Remarks to the Author):

The authors show a well-deserved response to my comments. At this state, i have no doubt in this work.

Response to Reviewers' Comments

We wish to express our heartfelt gratitude for the constructive feedback and insightful suggestions provided by the reviewers regarding our manuscript. We have given thorough consideration to these comments, conducted supplementary experiments, and made revisions to the manuscript accordingly. Please find below a comprehensive description of the changes, along with our point-by-point response to the reviewers' comments.

Review #1

Recommendation: Acceptance after minor revisions noted.

The manuscript titled "Tailoring grain boundary stability of Zn-Ti alloy for long-lasting aqueous Zinc batteries" has commendably incorporated the feedback from reviewers, resulting in a significantly improved overall content and detailed explanations. Notably, this article presents insightful findings on the zinc metal deposition mechanism on Zn-Ti alloy in relation to the nucleation potential of zinc metal and the formation of nucleation sites with Zn-Ti alloy at grain boundaries. These insights are substantiated through electrochemical analysis coupled with voltage profiles and further detailed analysis. The revised main figures and supplementary figures (S13 to S17), featuring computational modeling, electrochemical interpretation, and ex-situ microscope images of the surface morphology of zinc alloy electrodes and their interpretation, have further enriched the results. Thus, the authors should address several concerns below and revised the manuscript before publication in Nature Communications.

Response:

Thanks a lot for your thorough reading and insightful suggestions on this manuscript. We are honored that the improved electrochemical reversibility brought about by grain boundary engineering and the deeper exploration of Zn nucleation and growth mechanisms can obtain such a positive assessment. The manuscript has been substantially enhanced thanks to the feedback from your previous review. Furthermore, we highly value the reviewer's supplementary comments and have provided further explanations and necessary revisions.

1. Provide surface characterization data with CLSM or AFM imaging for both bare Zn and Zn-Ti alloy before the cycling process. This additional information will enhance the understanding of morphological changes by metallurgical engineering for Zn anode system.

Response:

Thanks for the reviewer’s constructive comment. In response, we have included top-view SEM images and CLSM imaging of both bare Zn and Zn-Ti alloy before the cycling process (**Supplementary Fig. 1b-c**). It is evident that the surfaces of both Zn anodes are relatively smooth, with low surface roughness. Specifically, the Zn-Ti alloy displays regular scratches, a result of the polishing procedure involved in the metallurgical process.

Changes:

Supplementary Figure 1. (a) Fabrication process of Zn-Ti alloy. (b) SEM images of pristine bare Zn and Zn-Ti alloy foils. (c) CLSM imaging of pristine morphologies on bare Zn and Zn-Ti alloy, and corresponding surface profiles extracted along the location marked. (d) CLSM images of bare Zn and

Zn-Ti alloy at higher magnification.

The bare Zn surface appears predominantly flat, with minor cracks, while the Zn-Ti alloy foil exhibits regular scratches, resulting from the polishing procedure in the metallurgical process. Nonetheless, the surfaces of both Zn anodes are relatively smooth, characterized by a low surface roughness measuring approximately 0.1 μm . (In the Supporting Information, Page R2)

2. Please include detailed information on the XPS analysis in the supplementary materials. This will aid readers in gaining a more thorough understanding of the characterization techniques employed in the research.

Response:

We thank the reviewer for carefully reviewing our manuscript and raising this point. In response to your valuable suggestions, we have incorporated a comprehensive discussion of the XPS results in the supplementary information. The adventitious carbon peak located at 284.8 eV was utilized for sample calibration. In **Supplementary Fig. 9e**, the S 2p XPS spectra clearly reveal that the peaks corresponding to SO_4^{2-} (168.7 eV) were detected on the surface of Zn anodes after cycling. Notably, the intensity of these peaks is significantly higher on bare Zn compared to Zn-Ti alloy. Meanwhile, the Zn 2p_{3/2} XPS spectra ascertain the existing forms of Zn species (**Supplementary Fig. 9d**), which exhibit two distinct peaks, namely Zn²⁺ (1023.3 eV) and Zn⁰ (1021.8 eV) (*Handbook of X-ray photoelectron spectroscopy*, Perkin-Elmer Corporation, 1992, 40, 221). It is worth noting that the Zn²⁺/Zn⁰ ratio is higher in bare Zn (23.5%/76.5%) than in Zn-Ti alloy (11.4%/88.6%). These findings suggest the formation of a sulfate by-product on the surface of Zn anodes after cycling, with a more pronounced effect observed for bare Zn. The combination of SEM (**Supplementary Fig. 9a-c**) and XRD (**Supplementary Fig. 9f**) results confirms the identification of the by-product resulting from parasitic reactions as zinc hydroxysulfate ($\text{Zn}_4\text{SO}_4(\text{OH})_6 \cdot x\text{H}_2\text{O}$).

Changes:

Supplementary Figure 9. Top-view SEM images of (a) Zn-Ti alloy and (b) bare Zn after 50 cycles at $1 \text{ mA cm}^{-2}/1 \text{ mAh cm}^{-2}$. (c) The EDS elemental mapping of deposits on bare Zn. (d) Zn 2p, (e) S 2p XPS spectra and (f) XRD patterns of Zn anodes after cycling.

The application of further XPS analysis unveils the surface chemical composition. The S 2p XPS spectra indicate the distinct SO_4^{2-} peaks at 168.7 eV on the surface of Zn anodes after cycling⁵, with significantly higher intensity observed on bare Zn. Meanwhile, the Zn 2p_{3/2} XPS spectra can be deconvoluted into two peaks at 1023.3 eV and 1021.8 eV, corresponding to Zn^{2+} and Zn^0 , respectively⁶. Notably, the proportion of Zn^{2+} for bare Zn (23.5%) exceeds that in the Zn-Ti alloy (11.4%). Combined with XRD results, the by-product resulting from parasitic reactions is identified as $\text{Zn}_4\text{SO}_4(\text{OH})_6 \cdot \text{H}_2\text{O}$ (PDF#39-0690), exhibiting a more pronounced effect observed for bare Zn. (In the Supporting Information, Page R7)

3. In Supplementary Figure 24, where the EIS results exhibit different shapes between Zn and Zn-Ti alloy, we suggest including an equivalent circuits in the supplementary materials to elucidate these variations. Additionally, consider explaining if the electrical resistivity was influenced by metallurgical

processes, and support this with I-V curves for clarity.

Response:

We would like to express our sincere gratitude for the valuable feedback provided by the reviewer. As illustrated in **Supplementary Fig. 24a-b**, the Nyquist plot of the bare Zn symmetric cell exhibits a characteristic semicircle in the high-frequency region, while the Zn-Ti alloy displays a high-frequency semicircle followed by a slanted line in the low-frequency region. Indeed, the distinctions in Nyquist plot shapes between pure Zn and Zn-based alloys have been extensively documented (**Fig. R1**). The more pronounced Warburg behavior in the Zn-Ti alloy indicates faster bulk diffusion within the alloy structure, while the elevated charge transfer resistance in bare Zn masks the Warburg resistance, a phenomenon supported by previous studies (*Nature Communications*, 2020, 11, 1634; *Nature Communications*, 2020, 11; 829, *Electrochimica Acta*, 2021, 368, 137626; *Journal of Energy Chemistry*, 2020, 51, 285-292). It is worth noting that as the temperature increases, the charge transfer resistance decreases, and the slope of the line also rises, indicating an increase in the diffusion coefficient (**Supplementary Fig. 24b**). This observation is consistent with the primary influence of temperature on diffusion. For bare Zn, as the temperature rises, the Warburg behavior becomes more distinguishable (**Supplementary Fig. 24a** and **Fig. R1e**). In summary, the improved charge transfer and enhanced diffusion collectively enhance the reaction kinetics of the Zn-Ti alloy, which is also evident in its superior rate performance in electrochemical tests (**Fig. 4b** and **Supplementary Fig. 23**).

Figure R1. Nyquist plots of symmetric cells using (a) Zn-Al alloy, (b) Zn-Cu alloy, (c) Zn-Sn alloy, (d) Zn-Se alloy and (e) Zn-P alloy.

In evaluating the impact of metallurgical process on electrical resistivity, we conducted current-voltage (I-V) measurements for both bare Zn and Zn-Ti alloy, as depicted in **Fig. R2**. The electrical resistivity of these two anodes is calculated as follows:

$$\rho = \left(\frac{U}{I}\right) * \left(\frac{S}{L}\right) \quad (1)$$

where U represents the applied voltage, I the response current, S the cross-sectional area of metallic foil and L the length of metallic foil. The calculated electrical resistivity for bare Zn and Zn-Ti alloy is found to be $1.51 \times 10^{-5} \Omega \cdot \text{m}$ and $1.76 \times 10^{-5} \Omega \cdot \text{m}$, respectively. The slightly higher electrical resistivity of Zn-based alloys can be attributed to lattice distortions caused by the different atomic lattice constants of alloying elements compared to Zn atoms. These lattice distortions strongly scatter electron motion, resulting in a reduction in electrical conductivity. In addition, the measured electrical resistivity of Zn foil is considerably higher than that of pure Zn (ca. $6.00 \times 10^{-8} \Omega \cdot \text{m}$). This disparity primarily results from the natural oxidation of Zn foil during storage, leading to the formation of a very thin, high-resistivity ZnO layer (which is difficult to identify via XRD) (*Journal of the*

Electrochemical Society, 1989, 136, 193C-203C; Nature Communications, 2022, 13, 7922).

Figure R2. I-V curves of bare Zn and Zn-Ti alloy foils.

The charge transfer processes occurring at the Zn/electrolyte interface are intricate and subject to debate, typically encompassing three pivotal stages: desolvation, adsorption, and electron transfer (*Advanced Energy Materials, 2022, 12, 2102707*). Given the excellent electrical conductivity of the metal substrate, the electron transfer barrier can be considered negligible (*Energy & Environmental Science, 2020, 13, 503-510*). Desolvation at the interface is often considered as the rate-determining step for deposition, governing the reaction kinetics throughout the process (*Langmuir, 2010, 26, 11538-11543; Journal of the American Chemical Society, 2019, 141, 9422-9429*). Moreover, desolvation takes place within the electrical double layer, a phenomenon inevitably influenced by the electrode characteristics (*Energy Storage Materials, 2023, 62, 102932*). An equivalent circuit is employed to simulate the electrochemical impedance spectroscopy results, where R_s stands for the ohmic resistance, R_{ct} the charge transfer resistance, CPE the constant phase element, and Z_w the Warburg resistance (**Supplementary Fig. 24c**). Of particular interest are the charge transfer resistances, which are subsequently utilized for calculations related to the activation energy of desolvation (**Supplementary Fig. 24d**).

Changes:

Supplementary Figure 24. Nyquist plots of symmetric cells using (a) bare Zn and (b) Zn-Ti alloy at different temperatures. (c) Typical equivalent circuit model. (d) The fitted curves and the corresponding activation energies calculated by Arrhenius equation.

The Nyquist plot exhibits a characteristic pattern featuring a semicircle in the high-frequency region and a slanted line in the low-frequency region. Randle's equivalent circuit is employed to fit the Nyquist plots, where R_s denotes the ohmic resistance, R_{ct} the charge transfer resistance, CPE the constant phase element, and Z_w the Warburg resistance. Notably, the Zn-Ti alloy demonstrates a more pronounced Warburg behavior, indicating faster bulk diffusion within the alloy structure^{12,13}. The charge transfer process at the interface primarily involves desolvation, adsorption and electron transfer¹⁴. Given the excellent electrical conductivity of metallic electrode, the electron transfer barrier can be considered negligible. Particularly, the desolvation of Zn^{2+} is widely recognized as the rate-determining step for Zn deposition, dictating the reaction kinetics throughout the process³. (In the Supporting Information, Page R23)

Review #2

I am impressed by the thorough revision of the manuscript and satisfactory answer to reviewers' comments. I am now convinced that this is a top-quality piece of work that should be accepted.

Response:

We sincerely appreciate the invaluable comments provided by the reviewer and the generous recommendation of our research.

Review #3

The authors show a well-deserved response to my comments. At this state, i have no doubt in this work.

Response:

We would like to express our gratitude to the reviewer for the meticulous review and positive feedback.